# Sparse intensity sampling for ultrafast full-field reconstruction in low-dimensional photonic systems
Egor Manuylovich ⓘ ✉

Phase-sensitive measurements usually utilize interferometric techniques to retrieve the optical phase. However, when the feature space of an electromagnetic field is inherently low dimensional, most field parameters can be extracted from intensity measurements only. However, even the fastest of the previously published intensity-only methods have too high a computational complexity to be applicable at high data rates and, most importantly, require data from CCD cameras, which are generally slow. This paper shows how a few intensity measurements taken from properly placed photodetectors can be used to reconstruct the complex-valued field fully in systems with low-dimensional feature space. The presented method allows full-field characterization in few-mode fibers and does not employ a reference beam. This result is 3 orders of magnitude faster than the fastest previously published result and uses 3 orders of magnitude fewer photodetectors, allowing retrieval of mode amplitudes and phases relative to the fundamental mode using only several photodetectors. This approach enables ultrafast applications of intensity-only mode decomposition method, including pulse-to-pulse laser beam characterization, providing an essential tool for experimental exploration of the modal dynamics in spatiotemporal modelocked systems. It can also be applied to ultrafast sensing in few-mode fibers and for coherent mode division-multiplexed receivers using quadratic detectors only.

Multimode optical systems are advancing wave physics through emergent phenomena such as spatial attractors[1,2] and multimode solitons[3,4], challenging existing theories and inviting interdisciplinary approaches from fields such as statistical mechanics and machine learning, thus providing ideal platforms for exploring high-dimensional nonlinear physics[5]. The applications of multimode fibers span from high-power lasers[6] to optical communications[7] and from astrophysics[8] to integrated photonic circuits[9].

The nature of multidimensional light behavior in such systems, including time, amplitude, phase, spatial, and polarization dynamics, makes characterization and exploration of such systems challenging. The high complexity of signal processing in multimode optical systems is one of the challenges in multimode technologies in telecommunications[9]. The necessity of mode demultiplexing dictates utilizing mode sorters together with employing interferometric measurements to retrieve complex-valued mode weights, hindering the rapid adoption of multimode fibers as multi-spatial-channel waveguides.

The full-field characterization of light in multimode systems imposes a challenge that echoes the broader issue of phase retrieval, which arises in a variety of areas of optics and photonics: from characterization of the

complex field of laser pulses to phase retrieval in X-ray crystallography and from imaging through scattering media to coherent diffraction imaging. This challenge is pivotal in scenarios where only the intensity of a wave can be measured, while the phase information, which is crucial for a full understanding of the wave's behavior, is lost. However, even when the phase can be measured via interferometric techniques, intensity-only methods for full-field characterization are highly beneficial due to their simplicity and ability to operate in situations where interferometric methods are impractical or impossible. These intensity-based approaches enable the reconstruction of phase information from intensity measurements alone, making them essential in fields where accurate wavefront characterization is crucial but direct phase measurement is challenging.

In multi-core, multimode and few-mode fibers, the problem of retrieving amplitudes and phases of eigenmodes is known as mode (modal) decomposition (MD)[10–12]. The standard and one of the most universal methods for wavefront reconstruction is digital holography[13,14]. However, this method is interferometric, which complicates the optical setup and requires a reference beam. Moreover, interferograms need to be measured with a CCD camera or a large array of photodiodes, which limits the

Aston Institute of Photonic Technologies, Aston University, Birmingham, B4 7ET, UK. ✉e-mail: e.manuylovich@aston.ac.uk

bandwidth of this approach because of the bandwidth of the CCD camera. These two drawbacks have been resolved separately for wavefront reconstruction in multimode fibers.

On the one hand, the intrinsic low dimensionality of the spatial distribution of the electromagnetic field in optical fibers, which can be represented as a linear combination of eigenmodes, led to approaches of physical mode separation, such as photonic lanterns[15] and multi-plane light conversion (MPLC)[16]. These approaches allow for the spatial separation of different eigenmodes. Due to physical decoupling into single-mode fibers, these approaches are widely used in optical communications, where they are utilized for mode-division-multiplexed (MDM) transmission links[7,17,18]. These approaches eliminate the need for large and slow arrays of photodetectors, such as CCD cameras, but still require local oscillators and optical hybrids for coherent detection of mode-encoded signals. The speed-of-light performance on mode decoupling became the deciding factor of the applicability of these devices in optical communications.

On the other hand, striving to remove the requirement for the reference beam while still fully characterizing the wavefront at an optical fiber led to the development of a wide variety of intensity-only mode decomposition approaches[11,19–21]. These methods usually involve the measurement of spatial intensity distribution at some measurement plane, while full wavefront information is retrieved via numerical methods. Despite being limited in speed by camera capabilities, the simplicity of intensity-only methods has led to their widespread use. In particular, intensity-only MD was applied as a diagnostic tool in high-power graded-index-fiber lasers and Raman amplifiers, demonstrating the detailed evolution of modal weights during the beam cleanup process[22]. It has also been utilized for sensing in few-mode fibers[23]. Another application is the study of the mode power distribution of ultrashort pulses in graded-index multimode fibers[24].

The intensity-only mode decomposition problem is essentially nonlinear, and the numerical methods used to solve it fall into two primary categories: iterative and noniterative approaches. Iterative algorithms include the classical Gerchberg–Saxton algorithm[11], line-search-based algorithm[25] and stochastic parallel gradient descent[26,27]. One of the recently published methods utilizes subsampled sampling for numerical MD[27]. The authors used SPDG, which limits the performance due to the iterative nature of this approach and makes it sensitive to the initial guess, while the lack of a systematic approach to the number of sensors and their placements resulted in a relatively high number of photodetectors being needed.

The family of noniterative approaches includes a variety of machine learning-based algorithms, primarily based on deep convolutional neural networks (CNNs)[19,28–30]. Deep unsupervised learning was also applied[31], which demonstrated MD in 10-mode fibers with low accuracy. There are several non-ML algorithms, including Fourier transform-based[20,32] and ptychography-based[33] algorithms. Combined approaches also exist that utilize noniterative MD for initial guessing and then refine the mode weights

using an iterative method, such as SPGD[34] or Grey Wolf Optimizer[35]. The typical speed of deep learning-based and other noniterative algorithms is tens of Hz[19,28]. Typically, these methods can decompose between 3 and 10 modes. A recent study demonstrated experimental MD in 8-mode fibers[36] with a speed of 65 ms per decomposition or approximately 15 Hz. We recently proposed a fast, noniterative algorithm[37,38]. This approach allowed for decomposition rates of up to hundreds of kHz in 3- and 5-mode fibers. In another study a similar method has also been recently applied to multi-core fibers[10]. Despite being several orders of magnitude faster than its competitors, the full beam intensity profile still needs to be measured.

The low speed and requirement of measuring large numbers of pixels limit the applicability of intensity-only methods in high-speed applications such as optical communications and high-speed sensing. To date, high-speed mode content retrieval relies solely on the physical separation of spatial modes, and methods such as MPLC dominate in MDM optical communications. A technique that allows for full-field characterization at GHz speed using only a few photodetectors and no reference beam is essential for multiple applications, including MDM optical communications and ultrafast sensing.

## Results

This paper introduces sparse sampling for pseudo coherent direct detection technique that utilizes only a few quadratic detectors to recover the full complex field in the measurement plane, including the amplitude distribution and phase distribution. We would like to emphasize that no reference beam is used, and the processing complexity in terms of the number of multiplications is 3 orders of magnitude lower than that of the current fastest published algorithm. The main advantage of this approach is that full wavefront reconstruction can be performed using only a few fast photodetectors, which essentially allows pseudo-coherent signal recovery via direct detection at telecommunication symbol rates.

The technique introduced in this paper allows the following: given some unknown field distribution from a fiber or another inherently low-dimensional system (see Fig. 1a) that corresponds to some intensity distribution (Fig. 1b), a few noisy photodetectors can be placed in a measurement plane (Fig. 1c) to measure the intensity of the electromagnetic field at corresponding points. Then, these few noisy and sparsely taken intensity measurements can be used to reconstruct the field distribution (Fig. 1d), which coincides with the true distribution up to a constant phase shift and complex conjugation.

This is done by recovering the complex-valued weights of the eigenmodes of the low-dimensional system. In what follows, we consider few-mode optical fibers with a step-index profile in the LP mode approximation. Still, a similar approach can be applied to other low-dimensional systems, such as planar photonic waveguides or photonic crystal fibers.

For an optical fiber that supports $M$ eigenmodes $\Psi_k$, the arbitrary field distribution at any cross section can be represented as a linear combination

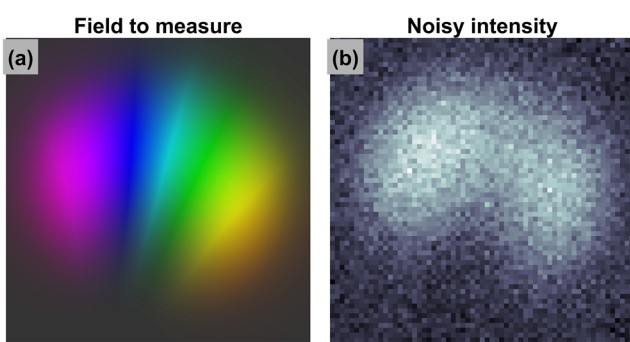
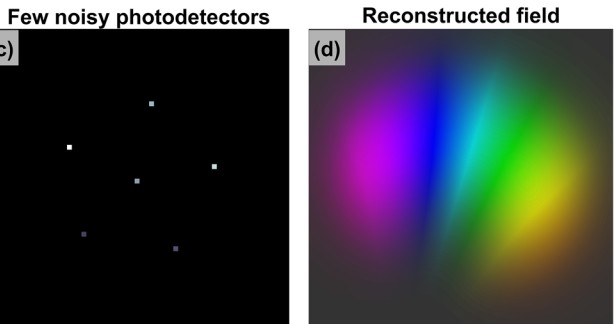

**Fig. 1 | Sparse sampling for complex field reconstruction. a** The original complex-valued field distribution, where intensity is represented by brightness (saturation-coded), and phase is represented by hue (color-coded); (**b**) noisy intensity distribution in the measurement plane, shown in grayscale; (**c**) sparsely sampled noisy intensity measurements, also displayed in grayscale, with noise levels corresponding to those in (**b**); (**d**) reconstructed complex-valued field, with intensity and phase represented as in (**a**). The phase values are mapped to a hue scale ranging from 0 to $2\pi$.

of these modes:

$$E(x,y) = \sum_{k=1}^{M} C_k \Psi_k(x,y) \quad (1)$$

These complex-valued coefficients $C_k = A_k \cdot exp(i\varphi_k)$, together with eigenmode field distributions $\Psi_k(x,y)$, fully describe the field in the transverse plane $(x,y)$. In telecommunications, when mode division multiplexing is used, the eigenmodes serve as separate channels, carrying encoded channel bit sequences, independently allowing for IQ modulation for each spatial mode. If one knows the field distribution $E(x,y)$, one can easily recover the complex weights of the modes by projecting this field into different spatial modes. However, to measure the complex field distribution directly, interferometric techniques such as off-axis digital holography are needed to recover the optical phase, which is usually elusive when simple quadratic detectors are used. The intensity or optical power, represented as the time-averaged squared absolute value of the electric field, can be easily measured:

$$I(x,y) = \left\langle |E(x,y)|^2 \right\rangle = \left\langle \left| \sum_{k=1}^{M} C_k \Psi_k(x,y) \right|^2 \right\rangle \quad (2)$$

The inference of complex mode weights from intensity measurements is a nonlinear and, to some extent, ambiguous task. The ambiguity arises from the fact that the intensity distribution remains unchanged when a constant phase shift is applied to the entire field or when a complex conjugate transformation is applied. Fortunately, only a few pieces of information are lost, and the majority can still be recovered using only intensity measurements. Traditionally, to disambiguate the phase shift, one would set $\varphi_1 = 0$. The method for disambiguating complex conjugate transformations depends on the measurement plane. This can be achieved for near-field measurements by restricting the phase of one of the $LP_{11}$ modes to lie within the range $[0, \pi]$. For far-field measurements, the phase should be within the range $[-\pi/2, \pi/2]$[38]. In the realm of optical communications, this implies that we are limited to amplitude modulation only for one of the modes. For another mode, only half of the IQ plane is accessible. All the other modes have full access to IQ modulation.

To recover mode weights from the intensity distribution $I(x,y)$, we rewrite Eq. 2 in the following form:

$$I(x,y) = \sum_{k=1}^{M} \sum_{j=1}^{M} C_k C_j^* \Psi_k(x,y) \Psi_j(x,y) \quad (3)$$

Note that here, we assume that $\Psi_k(x,y) = \Psi_k^*(x,y)$. We can do that for linearly polarized modes because we can always choose fiber modes to be real-valued and put all the complex-valued parts in the corresponding weight $C_k$. For example, when using a far-field measurement plane, the anti-central-symmetric modes can be turned into real-valued modes by "turning" them by the $\pi/2$ angle, i.e., by multiplying by $e^{i\pi/2}$ [38].

We can introduce new variables to linearize this system of equations:

$$I(x,y) = \sum_{i=1}^{L} S_i \Upsilon_i(x,y) \quad (4)$$

Here, the number $L$ of functions $\Upsilon_i(x,y)$ is equal to the number of different terms in Eq. 3 when we group complex conjugates, i.e., $L = M(M+1)/2$, while

$$S_i = \frac{C_k C_j^* + C_k^* C_j}{2}, \quad (5)$$

**Table 1 | Correspondence between eigenmode indices and intensity pattern indices**

|  | $j = 1$ | $j = 2$ | $j = 3$ | ... | $j = M$ |
|---|---|---|---|---|---|
| $k = 1$ | $\Upsilon_1$ |  |  |  |  |
| $k = 2$ | $\Upsilon_{M+1}$ | $\Upsilon_2$ |  |  |  |
| $k = 3$ | $\Upsilon_{M+2}$ | $\Upsilon_{2M}$ | $\Upsilon_3$ |  |  |
| ⋮ | ⋮ | ⋮ | ⋮ | ⋱ |  |
| $k = M$ | $\Upsilon_{2M-1}$ | $\Upsilon_{3M-3}$ | ... | ... | $\Upsilon_M$ |

This table shows how the indices of eigenmodes relate to those of the intensity pattern basis functions.

and

$$\begin{cases} \Upsilon_i(x,y) = 2\Psi_k(x,y)\Psi_j(x,y), & \text{if } k \neq j \\ \Upsilon_i(x,y) = \Psi_k^2(x,y), & \text{if } k = j \end{cases} \quad (6)$$

The $(k,j) \leftrightarrow i$ correspondence can be arbitrarily chosen, and we choose the one shown in Table 1.

The intensity distribution is now described as a linear combination of basis patterns $\Upsilon_i$ taken with some new weights $S_i$.

In typical practical applications of mode decomposition, the intensity distribution is usually measured with a CCD camera with $N$ pixels, which measures the intensity distribution across hundreds of thousands of individual points. In this case, the $(x,y)$ plane is discretized, and Eq. 4 breaks down into a system of linear equations, one equation per pixel:

$$I^{(n)} = \sum_{i=1}^{L} S_i \Upsilon_i^{(n)}, \quad n = 1..N, \quad (7)$$

We denote $\vec{\Upsilon}_i$ a vector composed of $N$ individual pixels for the $i$-th intensity pattern, $\vec{I}$ a vector composed of $N$ individual pixels of measured intensity distribution, $\vec{S}$ a vector of $L$ elements $(S_1, S_2, \ldots, S_L)^\top$, and matrix $\Upsilon$ is composed of $\vec{\Upsilon}_i$. Thus, the linear system of equations now reads:

$$\vec{I} = \Upsilon \vec{S}, \quad (8)$$

Now, the mode weights can be determined by solving the inverse linear part described in Eq. 8 using the Moore-Penrose pseudoinverse and then solving the inverse nonlinear part in Eq. 5 analytically. It should be noted that when the intensity distribution is measured with a camera, the linear part, Eq. 7, is strongly overdetermined. This linear part requires hundreds of thousands of multiplication operations and is the most time-consuming part of the mode decomposition algorithm.

Finding an efficient way to reduce or even eliminate the overdetermination of Eq. 7 while maintaining the accuracy of the solution would significantly reduce the computational complexity of mode decomposition while also paving the way for using fewer and faster photodetectors. Ideally, the camera could be replaced with a small number of telecom-grade photodetectors, which could be 9 orders of magnitude faster (tens of GHz vs. tens of Hz). At the same time, the computational complexity could be reduced by more than 3 orders of magnitude. In what follows, we investigate the key questions:

- What is the minimum number of equations required in Eq. 7 to fully characterize any observed intensity distribution?
- If we consider only a subset of Eq. 7, which coordinates $(x,y)$ should we select, or equivalently, which pixels should we choose?
- How can we estimate the quality of the chosen subset of pixels for the decomposition problem?
- How can complex-valued mode weights be reconstructed from these sparse intensity measurements?

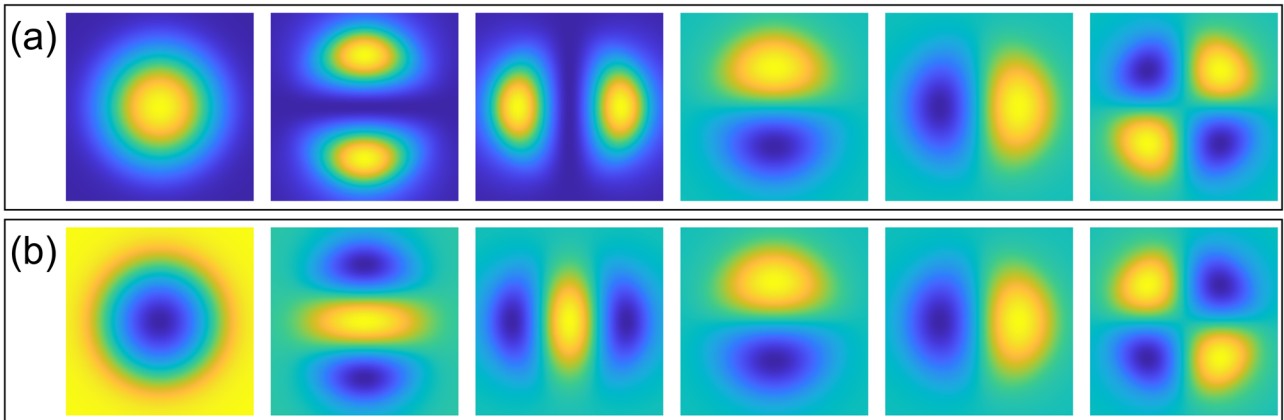

**Fig. 2 | Orthogonalized basis for 3-mode fiber.** Columns of intensity patterns $\Upsilon$ (**a**) and their orthogonalized versions $\Upsilon_\perp$ (**b**) for 3-mode fiber, reshaped in 2D patterns.

## Minimal number of measurements

We can answer the first question by recalling the fact that every field distribution in the transverse plane can be represented by a linear combination of eigenmodes, as described in Eq. 1, and that every intensity distribution can be represented as a linear combination of pairwise products of eigenmodes. Therefore, for $M$ modes, there are $L$ pairwise products, including self-products such as $\Psi_k \Psi_k$. Some of these intensity patterns may be degenerate and $\Upsilon$ may contain linearly dependent columns. Especially for a larger $M$, as the condition number of $\Upsilon$ increases exponentially with the number of modes[38]. The condition number of $\Upsilon$ is:

$$\kappa(\Upsilon) = \sigma_{\max}/\sigma_{\min} \tag{9}$$

Here, $\sigma_{\max}$ is the largest singular value, and $\sigma_{\min}$ is the smallest nonzero singular value of matrix $\Upsilon$.

One can be sure that the number of required measurements is not higher than $L$, which is significantly lower than the typical number of pixels $N$ used in mode decomposition approaches. Therefore, a few well-placed sensors are sufficient for determining the intensity pattern content of the intensity distribution and, consequently, complex weights of eigenmodes. In particular, for a 3-mode fiber, $L = M(M+1)/2 = 6$; therefore, only 6 measurements are sufficient for the full reconstruction of mode weights.

## Photodetector placements

To answer the second question, we follow the data-driven QR sensing paradigm with sparse sampling, which is basically a compressed sensing approach performed on a tailored basis[39]. First, we use QR factorization to orthogonalize the intensity basis:

$$\Upsilon = \Upsilon_\perp \mathbf{G}, \tag{10}$$

Here, the matrix $\Upsilon_\perp$ contains an orthogonalized basis and $\mathbf{G}$ describes the orthogonalization transformation. It has the same condition number as $\Upsilon$, and it is well-conditioned for a low number of modes. The individual column components of $\Upsilon$ and $\Upsilon_\perp$ reshaped into matrices, for a 3-mode fiber, are shown in Fig. 2.

As we have already determined the minimal number of measurements, the task is now to find the best positions for these measurements to robustly reconstruct the high-dimensional intensity distribution and mode weights.

The problem of optimal sensor placement can be mathematically formulated as follows. The intensity distribution has a low-rank representation in terms of intensity patterns $\Upsilon_\perp$:

$$\vec{I} = \Upsilon_\perp \vec{S_\perp}, \tag{11}$$

Here, $\vec{S_\perp}$ is a vector of intensity pattern weights in the orthogonalized basis $\Upsilon_\perp$, and the original vector $\vec{S}$ can be recovered by

$$\vec{S} = \mathbf{G}^{-1}\vec{S_\perp}. \tag{12}$$

The goal is to construct an optimized measurement matrix $\mathbf{P}$, effectively serving as a mask, consisting of rows of the identity matrix. The result of multiplying matrix $\mathbf{P}$ by matrix $\Upsilon_\perp$ is effectively reducing the number of rows in matrix $\Upsilon_\perp$ by selecting particular rows. This corresponds to individual photodetectors or individual pixels of a camera. This matrix is $p \times N$, where $p$ is significantly smaller than $N$:

$$\vec{I_p} = \mathbf{P}\vec{I} = \left(\mathbf{P}\Upsilon_\perp\right)\vec{S_\perp} = \Theta\vec{S_\perp}, \tag{13}$$

Here, $\vec{I_p}$ contains only $p$ elements, and $\Theta$ contains $p$ rows from $\Upsilon_\perp$. In this case, the original intensity weights $\vec{S}$ can be retrieved by

$$\vec{S} = \mathbf{G}^{-1}\Theta^\dagger \vec{I_p} \equiv \Xi\vec{I_p}, \tag{14}$$

Here, $\Theta^\dagger$ is the Moore–Penrose pseudoinverse of $\Theta$.

We need to find a subset of measured pixels $\vec{I_p}$ such that we can solve the inverse problem Eq. 13, i.e., $\Theta$; therefore, $\Xi$ are well conditioned.

Brute force optimization of matrix $\mathbf{P}$ to maximize the condition number of $\Theta$ leads to a combinatorial explosion, requiring a search over $\binom{N}{p}$ potential photodetector configurations. This becomes computationally challenging even for modest values of $N$. In this work, we follow empirical interpolation methods (EIMs) initially developed for partial differential equations[40], particularly the QR-pivoting discrete EIM[41], which provides near-optimal sensor placements adapted to a tailored basis[39].

The sensor positions we want to optimize correspond to the rows of the matrix $\Upsilon_\perp$; thus, we want to extract the most information about the intensity distribution by using as few rows of $\Upsilon_\perp$ as possible. To identify these row positions, we use column-pivoting QR factorization of matrix $\Upsilon_\perp^\top$, the transpose of $\Upsilon_\perp$.

Given a matrix $\Upsilon_\perp^\top$ representing measurements from all potential sensor locations, the pivoting QR factorization seeks to reorder the columns of $\Upsilon_\perp^\top$ such that the most "informative" columns (or sensors) are moved to the forefront. The measure of "informativeness" here is rooted in the magnitudes of the diagonal entries of the upper triangular matrix $\mathbf{R}$ resulting from the QR factorization. Sensors corresponding to larger diagonal entries in $\mathbf{R}$ are deemed more significant.

Another way of looking at it is that we select samples that exhibit the highest variability across the basis functions $\vec{\Upsilon_i}$ and those with the highest norm. These correspond to the largest columns in the $\mathbf{R}$ matrix from the column-pivoting QR factorization of $\Upsilon_\perp^\top$. These vectors resemble principal

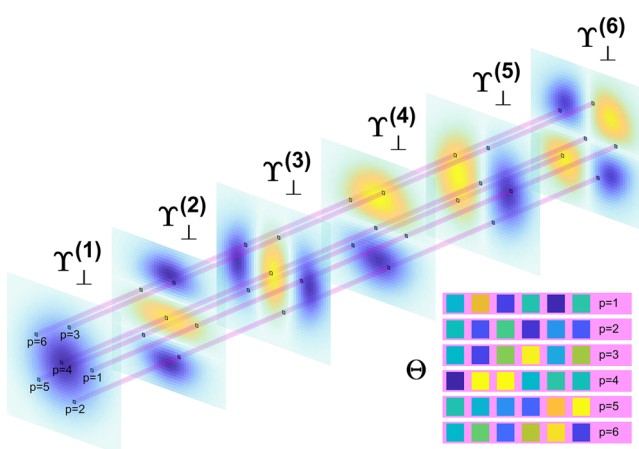

**Fig. 3 | Structure of the matrix $\Theta$ for 3-mode fiber.** Columns of orthogonalized intensity patterns $\Upsilon_\perp$, reshaped in 2D patterns, and matrix $\Theta$ composed of samples of the orthogonalized basis.

**Table 2 | Condition numbers for different configurations**

| $\Xi$ | 3 modes | | 5 modes | |
|---|---|---|---|---|
| | Near field | Far field | Near field | Far field |
| $(\mathbf{P}\Upsilon)^\dagger$ | 5.1 | 6.1 | 521 | 37.5 |
| $\mathbf{G}^{-1}(\mathbf{P}\Upsilon_\perp)^\dagger$ | 2.7 | 3.2 | 266 | 22.0 |
| $\Upsilon^\dagger$ | 2.7 | 2.7 | 202 | 18.0 |

The condition number of the decomposition matrix $\kappa(\Xi)$ depends on the number of modes, measurement plane, and content of $\Xi$: sparse measurements on pairwise mode products (first row), sparse measurements on the orthogonalized basis (second row), and full measurements on pairwise mode products (third row).

components, but they are chosen directly from the columns of the matrix $\Upsilon_\perp^\top$.

The structure of $\Theta$ for a 3-mode fiber and how this matrix is composed of the intensity basis are visualized in Fig. 3.

In practical terms, we construct the measurement matrix $\mathbf{P}$ by selecting the first $p$ columns corresponding to the largest $p$ diagonal values of the upper-triangular matrix $\mathbf{R}$ from the QR factorization of $\Upsilon_\perp^\top$. We can identify almost optimal sensor positions that enable the extraction of nearly maximal information using $p$ measurements, mitigating the need for a combinatorial search across all $\binom{N}{p}$ photodetector configurations.

Since we know that every intensity distribution in the measurement plane can be represented as a linear combination of columns of the matrix $\Upsilon_\perp$, we use

$$p = \text{rank}(\Upsilon_\perp). \tag{15}$$

For 3- and 5-mode fibers, $\Upsilon_\perp$ is full-rank in the far-field measurement plane, and the condition numbers of $\Upsilon_\perp$ equal 2.7 and 18, respectively. Therefore, we can simply use

$$p = L \tag{16}$$

or 6 and 15 photodetectors for 3- and 5-mode fibers, respectively.

It should be noted that the number of samples is lower than the typical number for compressed sensing, which is[42]:

$$p_{cs} \sim \mathcal{O}\left(L \cdot \log(N/L)\right). \tag{17}$$

The photodetector placements determined by pivoting QR factorization are also fundamentally different from random positions in compressed sensing, which must be incoherent with the universal basis. The optimized sensor placements, in our case, are data-driven and benefit from the use of a *tailored* basis $\Upsilon_\perp$ instead of the universal basis used in the compressed sensing approach. The high specificity of the data and the prior knowledge of the mode structure enable this opportunity.

**Quality of sparse photodetector placements**

Our goal is to find a matrix $\Xi$ such that it has the lowest condition number $\kappa(\Xi)$ to avoid noise amplification due to poorly conditioned linear transformation. One could apply the column pivoting QR factorization directly to $\Upsilon^\top$ without orthogonalization; therefore, $\mathbf{G}$ would be just an identity matrix, but we found that $\kappa(\Xi)$ is lower in the case of prior orthogonalization. Table 2 contains $\kappa(\Xi)$ for different cases using both orthogonalized and

non-orthogonalized bases and a comparison of these to the $\kappa(\Xi)$ of non-sparse measurements.

The third row represents the case in which we keep all $N$ measurements. Note that near-field measurements are slightly better for 3-mode fibers, and far-field measurements are far better than near-field measurements for 5-mode fibers. Similar behavior is observed for the 6-mode fibers. This is mainly caused by the better conditioning of $\Upsilon$ for the far-field measurements[38]. Additionally, using an orthogonalized basis $\Upsilon_\perp$ for finding sensor placements is better than using only $\Upsilon$. We believe this is caused by the fact that an orthogonalized basis inherently tends to spread out in the intensity pattern space, ensuring that the selected sensor locations capture a broader and more distinct range of speckle characteristics. In other words, by using $\Upsilon_\perp$, we can reduce the redundancy in the information collected by the sensors. This results in more effective sensing, as the orthogonal nature of the basis $\Upsilon_\perp$ ensures that the influence of each sensor location is distinct, leading to better conditioning $\Xi$.

The positions of the sensor placements for the orthogonalized basis, provided by the pivoting QR factorization, provide us with insight into what pattern they might follow. This pattern looks like a pentagon of sensors with one photodetector in the middle (see Fig. 4). The optimization of the photodetector placements in the classes of these circular patterns provides very similar results in terms of decomposition accuracy. However, knowing these circular patterns may be helpful in engineering this type of detector.

Therefore, we found that the determined sparse photodetector placements provide $\kappa(\Xi)$, comparable to the full-measurement approach.

**Recovering complex-valued mode weights from sparse intensity measurements**

Finally, once we determine the number of required photodetectors and their locations, we can recover $\vec{S}$ using Eq. 14 and then the complex-valued eigenmode weights $C_k$ using Eq. 5 and the simple numerical algorithm described in ref. 37 for the near-field measurement plane and in ref. 38 for the far-field measurement plane. In summary, the proposed technique is as follows:

1. The "intensity basis" is constructed from the pairwise products of eigenmodes $\Upsilon$
2. Orthogonalize it to obtain $\Upsilon_\perp$ and $\mathbf{G}$. Determine the required number of sensors as $p = \text{rank}(\Upsilon_\perp)$.
3. Locate the sensor positions using column-pivoting QR factorization
4. Compose $\Theta$ using the selected $p$ rows of the basis matrix $\Upsilon_\perp$
5. Measure $\vec{I_p}$ intensity values in the corresponding $p$ locations.
6. Calculate the vector $\vec{S}$ of the weights of pairwise products of the modes using Eq. 14.
7. Finally, calculate complex-valued mode weights by using Eq. 5. One requires only the first $3M - 3$ components of $\vec{S}$. For instance, the eigenmode amplitudes and phases for the near-field measurement plane are as follows:

$$\begin{cases} A_k = \sqrt{S_k}, & k = 1..M \\ \cos(\varphi_k) = \frac{S_{M-1+k}}{A_1 A_k}, & k = 2..M, \\ C_k = A_k \cdot \exp(\varphi_k) \end{cases} \tag{18}$$

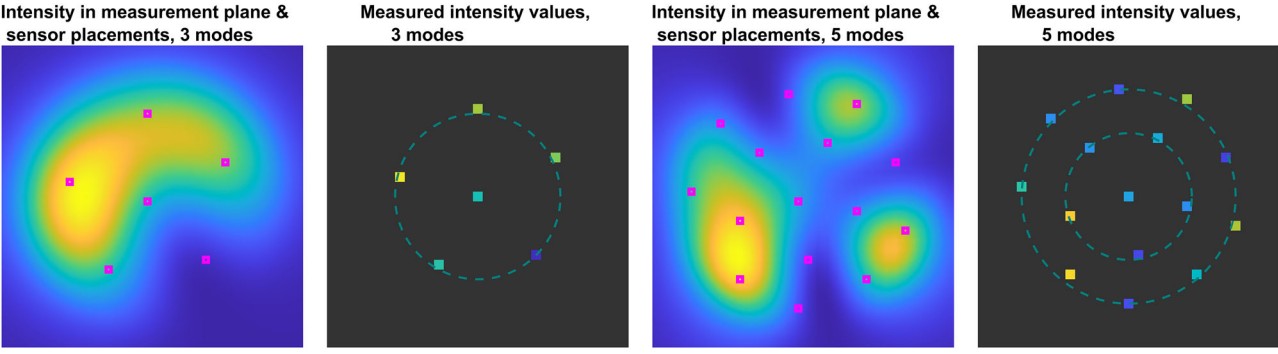

**Fig. 4 | Optimized sensor placement for sparse sampling.** Intensity distribution in the measured plane; photodetector placements, provided by column-pivoting QR factorization of transposed matrix of orthogonalized basis $\Upsilon_\perp^\top$; and measured intensities for 3- and 5-mode fibers. The data are simulated in the near-field measurement plane.

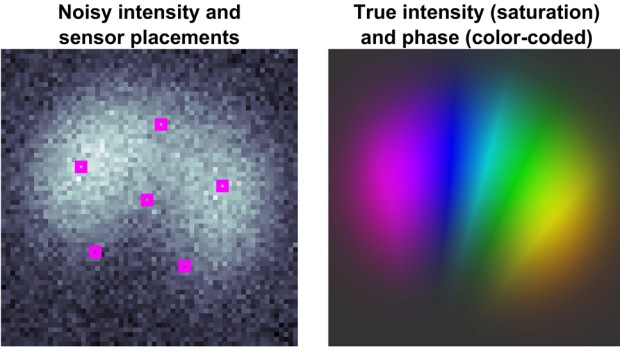

**Fig. 5 | Reconstruction from sparse measurements in 3-mode fiber.** In the near-field measurement plane, the figure depicts the noisy intensity distribution in grayscale, photodetector placements, the original intensity (saturation-coded for brightness) with phase (color-coded for hue), and the reconstructed intensity (saturation-coded for brightness) with phase (color-coded for hue). The true complex-valued mode weights and their reconstructed counterparts from noisy measurements are also shown. Phase values are mapped to a hue scale ranging from 0 to $2\pi$. The data are based on a 3-mode fiber with an SNR of 10 dB.

The exact phases from their cosine values can be recovered by extracting information on how these modes mix with the second mode[37]. We denote this nonlinear retrieval as $f(S_k)$:

$$C_k = f(S_k), \quad k = 1..M. \tag{19}$$

This notation is used in the "Methods" section.

The proposed method was tested for accuracy, noise resilience, and computational complexity. Figures 5 and 6 provide examples of corresponding sparse sampling mode decomposition in 3- and 5-mode fibers, respectively. These results were obtained through numerical simulations on $64 \times 64$ pixel grids. Later in Section 3, we present experimental results.

These figures show the noisy intensity distributions with photodetector placements (pink), true intensity (saturation-coded), and phase (color-coded) phase distributions and the corresponding retrieved intensity and phase distributions (saturation- and color-coded, respectively) obtained by the proposed method using only intensity measurements taken from the pink locations and true and retrieved complex-valued weights of eigenmodes.

**Decomposition accuracy and noise resilience**

To estimate the decomposition accuracy, we simulated a series of intensity distributions for some randomly assigned sets of weights and added Gaussian noise. Subsequently, mode decomposition was performed, followed by the calculation of retrieval errors. For each set of parameters, such as the SNR and number of modes, 10,000 random weight sets were used to determine the statistics and average the performance over a large dataset. Due to the nonlinear nature of the problem, the Gaussian noise in the input intensity data is transformed into non-Gaussian noise in the retrieved parameters. Given the enhanced kurtosis observed in the distributions of the retrieved data (see Fig. 7), the root mean square error (RMSE) metric may yield a distorted representation due to its high sensitivity to the relatively few outliers present. This sensitivity can disproportionately impact the metric's assessment, suggesting a less accurate model performance than is the case. The mean absolute error (MAE) was utilized to provide a more robust error quantification that minimizes the influence of outliers.

The MAE was calculated for retrieving mode weight scalar products $\epsilon_S$, mode amplitudes $\epsilon_A$, mode relative phases $\epsilon_\varphi$ and complex-valued weights $\epsilon_C$.

The formal definitions of these errors are given in Eq. 20:

$$\epsilon_S = \frac{\sum_{t=1}^{T} \sum_{i=1}^{L} \left| S_{t,i}^{true} - S_{t,i}^{retrieved} \right|}{T \cdot L}$$

$$\epsilon_A = \frac{\sum_{t=1}^{T} \sum_{k=1}^{M} \left| A_{t,k}^{true} - A_{t,k}^{retrieved} \right|}{T \cdot M}$$

$$\epsilon_\varphi = \frac{\sum_{t=1}^{T} \sum_{k=1}^{M} \left| \varphi_{t,k}^{true} - \varphi_{t,k}^{retrieved} \right|}{2\pi \cdot T \cdot M} \tag{20}$$

$$\epsilon_C = \frac{\sum_{t=1}^{T} \sum_{k=1}^{M} \left| C_{t,k}^{true} - C_{t,k}^{retrieved} \right|}{T \cdot M}$$

Here, $T$ is the number of sample intensity distributions, $L = M(M+1)/2$ is the number of pairwise mode products, and, as before, $M$ is the number of modes. These error metrics, depending on the SNR, are shown in Fig. 8.

For calculating the MAE at every point in Fig. 8, $T = 10^4$ intensity samples with a size of $64 \times 64$ pixels were used. Each of these intensity samples was simulated using a set of randomly assigned mode amplitudes $A$

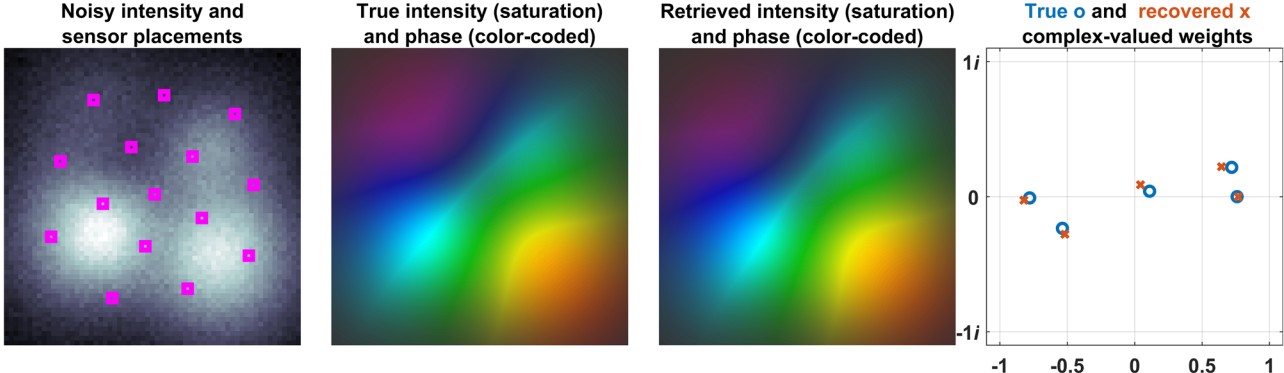

**Fig. 6 | Reconstruction from sparse measurements in 5-mode fiber.** In the near-field measurement plane, the figure depicts the noisy intensity distribution in grayscale, photodetector placements, the original intensity (saturation-coded for brightness) with phase (color-coded for hue), and the reconstructed intensity (saturation-coded for brightness) with phase (color-coded for hue). The true complex-valued mode weights and their reconstructed counterparts from noisy measurements are also shown. Phase values are mapped to a hue scale ranging from 0 to $2\pi$. The data are based on a 5-mode fiber with an SNR of 17 dB.

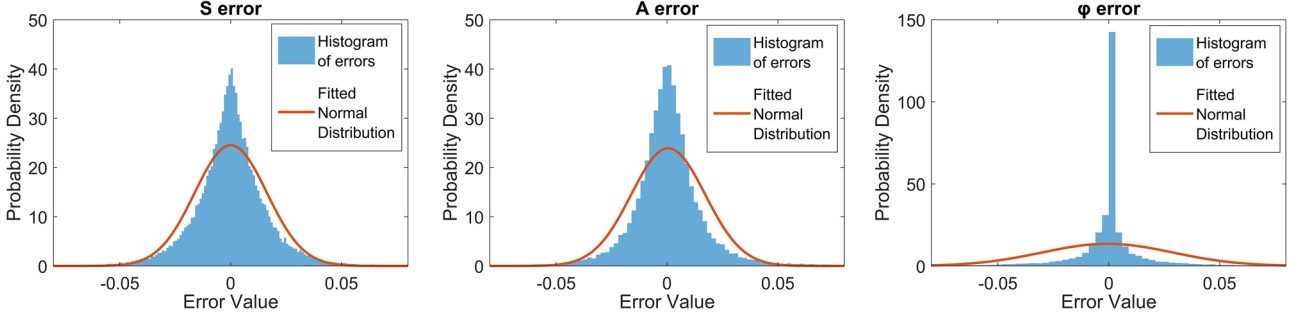

**Fig. 7 | Error distributions in mode decomposition.** Distributions of the errors in retrieving intensity basis coefficients $S$, mode amplitudes $A$ and mode phases $\varphi$ in a 3-mode fiber at SNR = 20 dB. The red curves show the best normal distribution fit for these errors. The poor fitness shows non-gaussian noise characteristics in the retrieved data. The high kurtosis and rare but distant outliers make the root mean square error less representative due to the high sensitivity to the outliers, so we used the mean absolute error as a more robust measure of the mean errors.

**Fig. 8 | Decomposition accuracy versus SNR.** The mean absolute errors $\epsilon_S$ of intensity basis coefficients, $\epsilon_A$ of mode amplitudes, $\epsilon_\varphi$ of mode phases and $\epsilon_C$ of complex-valued mode weigths, depending on the SNR of the intensity signal for 3- and 5-mode fibers.

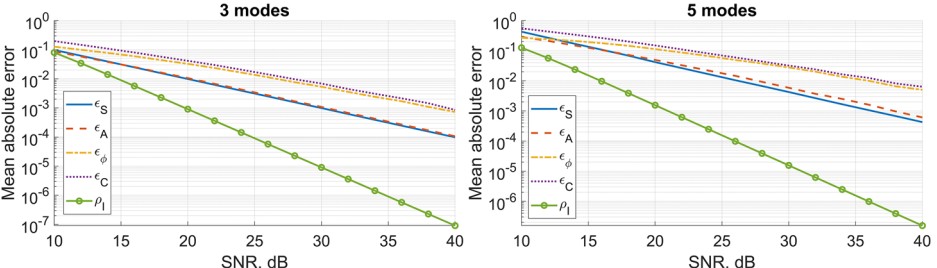

and phases φ. Mode amplitudes were taken from a uniform distribution $A_{t,k}^{true} \sim \mathcal{U}(0,1)$. The second mode phase was taken from a uniform distribution $\varphi_{t,2}^{true} \sim \mathcal{U}(0,\pi)$ for near-field simulations in a 3-mode fiber and from a uniform distribution $\varphi_{t,2}^{true} \sim \mathcal{U}(-\pi/2, \pi/2)$ for far-field simulations in a 5-mode fiber. This was done to ensure one-to-one correspondence between the mode weight sets and intensity distributions. The higher mode phases were taken from a uniform distribution $\varphi_{t,k>2}^{true} \sim \mathcal{U}(0, 2\pi)$. For each of the $T = 10^4$ generated intensity distributions, 10 noisy intensity distributions were generated by adding Gaussian noise with amplitudes corresponding to SNR = 10, 12, 14, 16, 18, 20, 25, 30, 35, and 40 dB. For each of these noisy patterns, the described mode decomposition was applied, and the retrieved weights $S_{t,i}^{retrieved}$, $A_{t,k}^{retrieved}$, $A_{t,k}^{retrieved}$, and $C_{t,k}^{retrieved}$ were used to calculate error metrics via Eq. 20. Figure 8 also shows intensity correlation distance (see Eq. 21) for the same parameters.

The proposed technique allows retrieving mode amplitudes in a 3-mode fiber with $\epsilon_A$ less than 0.1 when the SNR is as low as 10 dB. To recover complex-valued weights in a 3-mode fiber with $\epsilon_C < 0.1$, an SNR of 16 dB is needed. For a 5-mode fiber, an SNR of 16 dB is required to recover all 5-mode amplitudes with an accuracy better than $\epsilon_A < 0.1$, while an SNR better than 22.5 dB is needed to recover complex-valued weights with an accuracy better than $\epsilon_C < 0.1$.

**Computational complexity and comparison to the other intensity-only methods**

The computational complexity of the proposed technique was analyzed in terms of the number of multiplications $Q$ needed to perform a single full-field characterization. The proposed approach was compared to other noniterative intensity-only mode decomposition approaches, particularly

the previous fastest MD method and one of the popular CNN-based approaches[19]. A comparison with iterative approaches was deliberately avoided because the computational complexity can depend substantially on hyperparameters such as the number of iterations and stopping criteria. For the developed method, the number of photodetectors increases as the number of modes squared: $p \propto L = M(M + 1)/2$. To fully reconstruct $\vec{S}$ using the measured intensities, one needs to multiply these measured values by the matrix $\Xi$. However, for reconstructing mode weights $C_k$, only the first $3M - 3$ elements of $\vec{S}$ are utilized (see Eq. 18): first $M$ are used to determine mode amplitudes, next $M - 1$ are used to find cosine values of higher mode phases and the last $M - 2$ are required to determine the actual phases using their cosine values and their mixing with the second mode. This gives the computational complexity of the linear part $Q_{lin} = 6 \times 6 = 36$ multiplications for a 3-mode fiber and $Q_{lin} = 12 \times 15 = 180$ multiplications for a 5-mode fiber. The nonlinear part requires $Q_{nln} = 4(M - 1)$ multiplications[38] if the square root function and inverse trigonometric functions are calculated by using lookup tables. This gives the total required number of multiplications of $Q_{tot} = 44$ for 3-mode fibers and $Q_{tot} = 196$ for a 5-mode fiber.

In comparison, the previous fastest intensity-only mode decomposition approach provides a complexity of $Q_{FMD} = (3N + 4)(M - 1)$ for intensity distributions with $N$ pixels and consisting of $M$ modes. For a typical image size of $128 \times 128$ pixels, this yields 98312 multiplications, or 2234 times lower computational complexity for the proposed technique. For 5-mode fibers, the complexity reduction coefficient is 1003. For another popular approach, based on convolutional neural networks[19], the number of multiplications required is $Q_{CNN} \approx N \cdot \left(4478 + 2^{M-1} \cdot (2M - 1)\right)$[38], which yields 73,695,232 multiplications on $128 \times 128$ images for 3-mode fibers. This corresponds to 1,674,891 times lower computational complexity of the proposed technique compared to the CNN-based approach. For 5-mode fibers, the CNN-based approach requires 75,726,848 multiplications for $128 \times 128$ images or 386,361 times greater computational complexity than the proposed method.

## Discussion

A method for retrieving eigenmodes' complex-valued weights in few-mode fibers by using only a few quadratic photodetectors is proposed. The developed technique requires only 6 PDs for 3-mode fibers and only 15 PDs for 5-mode fibers to retrieve complex-valued mode weights. This is at least 3 orders of magnitude less than what is typically used in other noniterative methods. The computational complexity is shown to be more than 3 orders of magnitude lower than that of the current fastest intensity-only mode decomposition approach.

A data-driven approach was used to find sparse sensor placements, avoiding the astronomical combinatorial search through the sparse placement parameter space. It should be emphasized that the proposed method can be applied to full-field characterization using only a few intensity measurements in any photonic system in which the feature space of an electromagnetic field is highly degenerate or inherently low-dimensional, i.e., in which any field distribution can be represented as a linear combination of few modes. For example, a planar waveguide, a resonator, etc., can be used. The approach remains the same, and only the corresponding eigenmodes $\Psi_k$ should be used instead of LP modes, which were discussed in this paper.

The proposed approach shows good accuracy and speed for 3- and 5-mode fibers, even in the case of noisy measurements. The proposed method faces challenges with mode decomposition in fibers supporting six or more modes due to symmetries in the intensity patterns. These symmetries increase the condition number $\kappa(\Xi)$, leading to significant noise amplification during the inference stage. For example, in a six-mode fiber in the far-field plane, the condition number of the full matrix $\Upsilon$ is $\kappa(\Upsilon) \approx 200$, which corresponds to a noise amplification of approximately 23 dB. This results in substantial reconstruction errors, rendering the method impractical for fibers with a higher number of modes. However, compared to the previous fastest intensity-only method, this approach allows mode

decomposition for the same number of modes but with 3 orders of magnitude lower computational complexity. Compared to a popular non-iterative CNN-based approach, this approach allows full-field characterization in 5-mode fiber maximum versus 8 modes for CNN-based MD but is 6–7.5 orders of magnitude faster.

Unlike mode demultiplexing approaches such as photonic lanterns and multi-plane light conversion, which rely on physical spatial mode separation, this method offers the advantage of not requiring a complicated 3D photonic device for the physical separation of modes and eliminating the need for a reference beam to determine mode phases, as these phases are determined relative to the fundamental mode. However, this approach has several limitations, including the inability to determine the phase of the fundamental mode and the inherent limitation when applied to modes that are complex conjugates of each other. In such cases, intensity measurements are insufficient to distinguish between these modes, as the intensity patterns $\Upsilon_i$ corresponding to these modes are identical and, therefore, matrix $\Upsilon$ is singular. A notable example of this limitation arises with orbital angular momentum modes with rotational mode number $l = \pm 1, 2, \ldots$ as these modes exhibit identical intensity patterns in both the near-field and far-field, making it impossible to determine the direction of phase rotation (i.e., clockwise or counterclockwise) from intensity measurements alone. Since their field distributions are complex conjugates, distinguishing between them requires either additional phase information or temporal field tracking as mode coupling changes. Additionally, when eigenmodes are not linearly polarized and are not complex conjugates of themselves, i.e., if the condition of $\Psi_k(x, y) = \Psi_k^*(x, y)$ does not hold, then the number of linearly independent intensity patterns and, therefore, the number of photodetectors required increases from $L = M(M + 1)/2$ to $L = M^2$, which corresponds to 9 PDs for a 3-mode system instead of 6. Another important aspect is that when polarization multiplexing is used, and different polarizations are separated, such as with a PBS, wavefront reconstruction can be applied to each polarization component individually. However, the phase shift between the polarization components cannot be retrieved from intensity measurements of individual polarizations alone. Additional intensity measurements before the PBS are required to recover this phase shift. However, we note that once the mode weights in both polarizations are known, an additional intensity measurement (this time before polarization separation) enables the recovery of the polarization state's ellipticity. This allows for determining the phase shift between different polarization components, albeit up to $\mod(\pi)$. A detailed analysis and derivation of the inter-polarization phase shift are beyond the scope of this work.

Nevertheless, the simplicity of the proposed approach in not requiring complex photonic devices or a reference beam remains a significant advantage.

The feasibility of the approach was experimentally demonstrated with a 3-mode fiber using only 6-point intensity measurements. The computational complexity of the proposed technique is shown to be of only $Q = 44$ multiplications for 3-mode fibers per complete wavefront characterization and only $Q = 196$ multiplications for 5-mode fibers. This is another giant leap forward of more than 3 orders of magnitude in speed compared to MD using the fastest intensity-only mode decomposition technique to date[37,38]. 

Considering the low number of sparsely distributed intensity measurement locations, it can be claimed that only a few fast photodetectors can be utilized for measuring mode amplitudes and phases relative to the fundamental mode. These photodetectors can have a bandwidth of tens of GHz, which is 9 orders faster than that of a typical CCD camera previously used for MD. This unique feature, together with its 3 orders of magnitude lower computational complexity, validates the opening venue for ultrafast applications of this approach, including the facilitation of pulse-to-pulse beam full characterization of spatiotemporally modelocked lasers, providing an essential tool for experimental exploration of the fundamental scientific problem of modal dynamics in such systems. It can also be applied to ultrafast sensing in few-mode fibers, ultrafast phase-contrast imaging, and utilizing a few quadratic detectors as pseudo-coherent receivers in mode division multiplexing optical communications.

## Methods

To empirically validate the results, we conducted an experiment utilizing a piece of Hi-1060 fiber in conjunction with a 650 nm laser diode. The V number for this setup was determined to be approximately 3.5, corresponding to the support of three spatial modes: $LP_{01}$, $LP_{11o}$, and $LP_{11e}$. We used a 4-f imaging scheme with lenses with focal lengths of 4.51 mm (Thorlabs C230TME) and 300 mm (Thorlabs LB1779-B), which corresponds to a magnification factor of 66.5. We used a film polarizer (Thorlabs LPVIS050-MP2) in the focal plane to select a single linear polarization. We projected the magnified near-field pattern on a CCD camera (Ximea mq013rg-e2). The region 128 × 128 pixels in size that contained the beam was used for processing. The cropped image from the camera was used for alignment and finding basis mode functions that correspond to the measured intensity distributions. The positions of the 6 sensors used for mode decomposition were found using column-pivoting QR decomposition. Mode decomposition was performed on the chosen 6 pixels only. We collected a dataset of 600 near-field intensity distributions by using a motorized fiber polarization controller (Thorlabs MPC320) as a polarization scrambler and mode mixer. The whole measured dataset of intensity distributions was randomly split at an 80/20 ratio. The larger part was used to find the coordinates of the fiber center relative to the measured intensity distribution as well as the diameter of the fiber (in pixels) that corresponds to the measured intensity distributions. The preprocessing stage included subtracting the noise average value that was determined as a maximum on a kernel density estimation (KDE) function applied to intensity distributions. The peak at low intensities (see Fig. 9) corresponds to low-level pixels outside of the mean beam area.

Two main reasons for this peak are the imperfections of the optical system and the light scattered from the elements of the optical system. The decomposition was performed using only 6 values of the measured intensity, corresponding to 6 individual photodetectors. The background noise level can be measured well outside of the main beam area using a single additional photodetector; therefore, even an imperfect optical system can be used for real use-case scenarios.

In practical applications, background measurements can be effectively achieved by employing an additional photodetector outside the main beam area, while parasitic scattering can be reduced, and overall noise minimization can be achieved through a refined experimental scheme that includes coated optical surfaces and lower noise photodetectors.

After the initial preprocessing, the center of the beam and the corresponding core radius were determined using numerical optimization via the Nelder–Mead algorithm. The three parameters, which are the core image radius and the x and y coordinates of the center of the core image, were iteratively updated until the minimal mean correlation distance was achieved across all of the training samples.

The correlation distance $d = 1 - \rho$ is simply a measure of how far the correlation is from a perfect correlation. We used the most common intensity correlation function[25] as a measure of the accuracy of mode decomposition:

$$
\begin{aligned}
d\left(I_{\text{meas}}, I_{\text{retr}}\right) &= 1 - \rho\left(I_{\text{meas}}, I_{\text{retr}}\right) \\
\rho\left(I_{\text{meas}}, I_{\text{retr}}\right) &= \left|\frac{\iint \Delta I_{\text{meas}} \Delta I_{\text{retr}}\, dx dy}{\sqrt{\iint \Delta I_{\text{meas}}^2 dx dy \cdot \iint \Delta I_{\text{retr}}^2 dx dy}}\right| \\
\Delta I_{\text{meas}}(x, y) &= I_{\text{meas}}(x, y) - \overline{I_{\text{meas}}} \\
\Delta I_{\text{retr}}(x, y) &= I_{\text{retr}}(x, y) - \overline{I_{\text{retr}}}
\end{aligned}
\tag{21}
$$

Here, $I_{\text{meas}}$ and $I_{\text{retr}}$ denote the corresponding measured and retrieved intensity distributions, respectively, and $\overline{I}$ denotes the average intensity over the whole measured 128 × 128 pixel area.

$$
\begin{cases}
r, x_0, y_0 = \underset{r_0, x_0, y_0}{\arg\min}\left[\sum_i d\left(I_{\text{meas}}^{(t)}, I_{\text{retr}}^{(t)}\right)\right] \\
I_{\text{retr}}^{(t)} = \left\langle \left|\sum_{i=1}^{M} C_k^{(t)} \Psi_k\left(r, x_s, y_s\right)\right|^2 \right\rangle \\
C_k^{(t)} = f\left(S_k^{(t)}\right) = f\left(\Xi \vec{I_p}^{(t)}\right)
\end{cases}
\tag{22}
$$

Here, $\vec{I_p}^{(t)}$ are the 6 numbers corresponding to the measured intensities for the $t$th sample from the training dataset, $C_k^{(t)}$ are the retrieved mode weights corresponding to the $t$th sample image, and $\Xi$ is determined for the current basis $\Psi_k\left(r, x_0, y_0\right)$ using Eqs. 6, 10, and 13–14.

**Fig. 9 | Noise background in experimental measurements.** Intensity distribution on a noisy background (insert on **a**) highlighting the "noisy pedestal" outside the main beam area, and KDE near the lower intensity limit (**b**). The estimated noise level in the experiment is 18.5. The insert on the right-hand side shows the KDE for the whole range. The maximum of the noisy background peak is considered the mean of the noise distribution, and this value is subtracted from the whole image in the preprocessing stage.

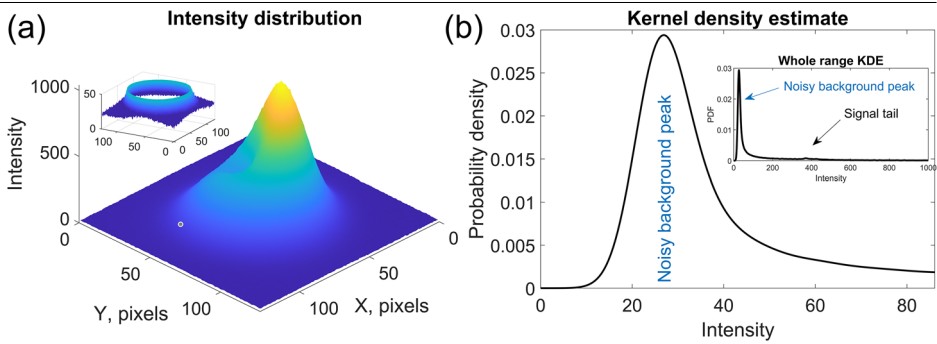

**Fig. 10 | Correlation distances across datasets.** Correlation distances on a log10 scale for training and testing data samples.

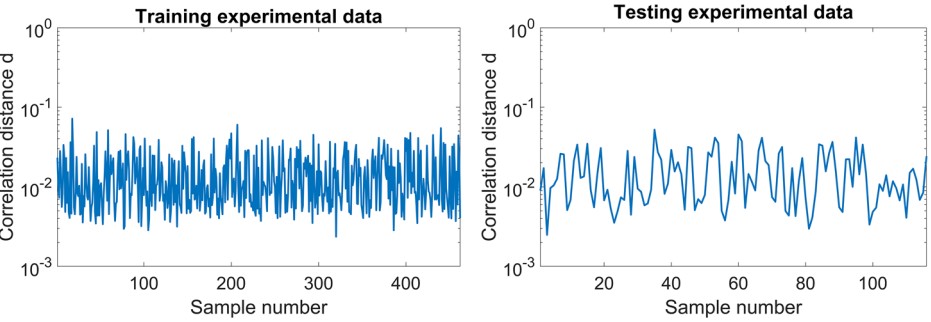

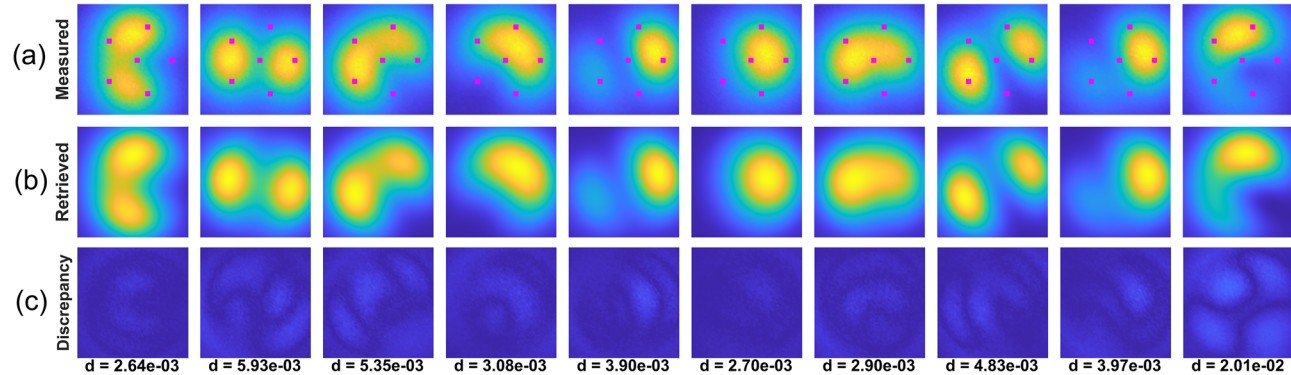

**Fig. 11 | Experimental reconstructions from sparse sampling.** Some of the measured samples with the positions of the intensity pixels used for mode decomposition (**a**), corresponding retrieved intensity distributions using the decomposed complex-valued mode weights $C_k$, Eq. 2 and the determined basis of eigenmodes $\Psi_k(r, x_0, y_0)$ (**b**), the discrepancy between them (lower row) and corresponding correlation distances (**c**).

We find such a set of parameters $(r, x_0, y_0)$ of basis $\Psi_k$ that when we apply the mode decomposition procedure to the training dataset, the average correlation distance between the measured and retrieved intensity distributions across the entire training dataset is minimized. Therefore, we minimize the average error on the training dataset. It should be emphasized that the basis $\Psi_k$ is the same for all the measured samples.

Figure 10 shows how the $\log_{10}$ of the training and testing correlation distances $d$ vary across the datasets. These two parts were not significantly different, even though the test part was not used for the basis determination. Therefore, we conclude that we properly identified the basis functions for our experimental setup.

Some of the experimental samples with retrieved intensity distributions are shown in Fig. 11. The positions of the "sensors" (i.e., the pixels used as sparse sensors) are depicted in pink.

## Data availability
The raw experimental dataset and the basic implementation of the proposed approach for near-field measurements are available at https://github.com/egor-manu/sparse-intensity-sampling-ultrafast-full-field-reconstruction.

## Code availability
The code supporting the implementation of the proposed sparse intensity sampling method for near-field measurements is openly available at https://doi.org/10.5281/zenodo.15097632.

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

## Acknowledgements
E.M. acknowledges the support of the EPSRC project EP/W002868/1 and the EU Horizon Europe Grant Agreement 101169118 HORIZON-MSCA-2023-DN-01-01.

## Author contributions
Dr. Egor Manuylovich conceived and developed the algorithm, tested it, developed the experimental scheme, performed the experiment, processed the experimental data, and wrote the paper.

## Competing interests
The author declares no competing interests.
