## [Transparent Peer Review file · Communications Physics]

Sparse intensity sampling for ultrafast full-field reconstruction in low-dimensional photonic systems

Corresponding Author: Dr Egor Manuylovich

Version 0:

Reviewer comments:

Reviewer #1

(Remarks to the Author)

Sparse intensity sampling for ultrafast full-field reconstruction in low-2 dimensional photonic systems

In this article the authors demonstrate a method to reconstruct the complex field transmitted by low mode optical fibers with only intensity measurements without the need of an interferometer. The intensities are measured at a few points (sparse sampling) arranged in circular patterns.

The article is similar to a previous article on "Intensity-only-measurement mode decomposition in few-mode fibers" (E. Manuylovich et al, Opt. Exp. 29, 36769, (2021)) and the main difference is that instead of using the full image captured by the camera now it is possible to only use the measurements in a few pixels in the camera.

The main advantage of this method is that, in principle, one could build an array of fast photodiodes placed at specific coordinates in order to achieve the rates of GHz. The disadvantage is that you have to calculate those optimal positions and then build the fast photodiode array at exactly those positions that only work for a specific fiber.

-In the abstract and the conclusion you claim that your result is 3 orders of magnitude faster than the fastest previously published result. The Opt. Exp. Paper (also claims to outperform the fastest), the mean decomposition time is 52 microseconds (Caption of Fig. 15). I understand that the 3 orders of magnitude refer to the number of multiplications, which might not be a 3 orders of magnitude in the processing time?

-Figures 1, 5 and 6 show complex images (intensity and phase), but there is no colorbar or explanation of the colorcode. Also, what is the size of the images in terms of pixels in the camera (and pixel size)?

-The experimental results of Fig. 11 only show the intensity distributions. What about the phase distributions?

-Also, it seems that right now your experiment is limited to a maximum of 5 modes. Is this also true for simulations? You mention the symmetries in the intensity patterns, which would appear to be a fundamental limitation? Or what is the mode number limit for your simulations?

Reviewer #2

(Remarks to the Author)

Overall, I think the technique could have some utility, but I don't think it would be particularly useful for use directly in mode division multiplexing, which appears to be the emphasis in the paper. I believe some of the other areas mentioned in the paper such as general beam characterization and few-mode fiber sensing would be more promising.

I'll start here by listing some problems I see for using the system directly in mode division multiplexing, I'll get these negative points out of the way first as there's other potentially better applications I can think of which I'll discuss in a moment.

Downsides of the method for direct use as mode demultiplexer...

- It's quite lossy, the photodetectors only sample a small portion of the beam.
- It can't distinguish between conjugate versions of the beam. The paper mentions limiting the modulation range to get around this, but presumably at the end of a fiber, with unknown mode coupling, you could end up with a scenario along the

lines of having to distinguish between an OAM +1 mode, and the conjugate OAM -1, and have no ability to tell the difference. Also, limiting the modulation range would limit the information capacity of the link, and would probably not be very popular unless there's a significant upside that comes along with it.

- How could your technique be used along with polarization? For mode division multiplexing you would need the ability to decipher the phase shift between polarization components. It would typically also be desirable to use wavelength division multiplexing as well.

- I suspect your scheme cannot reconstruct multiple fields at once? You've shown nice examples of recovering a single field, but for mode division multiplexing, the output beam would consist of multiple fields corresponding with multiple independent information channels, all with their own amplitude and phase modulations over the top. To be true mode demultiplexer you would have to be able to recover them all at once.

There has also been work on 'High-dimensional Stokes receivers' by groups such as William Shieh at University of Melbourne and Joseph Kahn at University of Stanford

<https://doi.org/10.1364/OL.44.002065>

That can reconstruct multiple channels at once, with no local oscillator (external reference), at telecommunications rates (GHz). These use more photodetectors than your approach, but these can also unambiguously recover multiple channels.

Potentially better application could be few-mode fiber sensing?

It might be more useful, not so much for telecoms, but as a general characterization or sensing tool. Perhaps in some application like sensing where you want to measure the drift of the modal content over time to discern some property (such as strain or temperature?), that drift might even help to distinguish between ambiguities that you can't otherwise resolve. For example, the difference between an OAM +1 mode, and an OAM -1 mode might be clear due to the modal content that preceded it in time? In that case you might only want a few modes, and you might want the high-speed aspect without going to a full-on coherent detection scheme? The simplicity of a few well placed photodetectors and high-speed reconstruction could be useful there. Perhaps it would also be useful for scenarios at wavelength where coherent detection isn't convenient or possible? i.e. not at 1310/1550nm. Just as you've done in this work where you've demonstrated at 650nm.

This is just an observation, but it looks like the results of Fig. 3, are similar/same to those found for attempting to find the optimal placing of spot couplers for N=6 fibers in coherent MDM. The mode basis looks the same, and the spot placement looks the same.

e.g. <https://ieeexplore.ieee.org/abstract/document/6317139>

2.3 : I was confused by the difference between near-field and far-field measurements. I don't know if you mention anywhere what type of fiber/modes you're working with? Is it step-index? If it's step-index I understand how there would be a difference between near and far-field, for graded-index (parabolic), I'd expect the near and far-fields to be identical. Later on in Section 3, you mention that it's step-index, but mentioning that somewhere earlier for the simulations of Section 2 would be helpful.

Am I correct in assuming that tests such as Fig. 5 are generated numerically (i.e. these aren't measured on real cameras etc?), it seems from later in the document that it is simulated, but it would be good to clarify earlier. And then in Section 3, it becomes experimental results?

Minor : the intensity plots of Figure 9 might be better as simply traditional 2D plots rather than 3D, it'd be easier to see the features in 2D.

In your 4. Discussion and Conclusion, you state "in any photonic system in which the feature space of an electromagnetic field is highly degenerate or inherently low-dimensional", I think you should change that to be "in other photonic systems", i.e. I don't think it'd work for "any" low-order photonic system. For example, an array of single mode waveguides? E.g. 3 single-mode waveguides is a 3 mode system, but I don't think your system could handle that scenario because the individual single mode waveguides don't occupy any of the same space, so you can't measure their interference. Although perhaps it could still work if you look at it in the far-field where the waveguides can interfere?

Minor (e.g. typos)...

Abstract : "have too high a computational complexity to be applicable at high {word missing, presumably 'speed'}?"

Typo, Fig 4 "lnIntensity"

Reviewer #3

(Remarks to the Author)

Version 1:

Reviewer comments:

Reviewer #1

(Remarks to the Author)

The authors answered my comments and clarified the points that I raised.

Reviewer #2

(Remarks to the Author)

2. "The proposed basic implementation is indeed unable to distinguish between conjugate fields. However, this limitation can be addressed with a simple modification: taking an additional intensity measurement in another plane."

I don't think this is the case specifically for OAM +/-1 modes. My understanding is you could use the two plane approach for almost all fields, except for a few specific cases where it fails. Namely, scenarios where two modes (e.g. OAM+ and OAM-) both have identical near-fields and far-fields. You can't tell the difference between those two orthogonal modes, because they have the property of looking the same in both planes. i.e. you'll see a donut in the near-field, and a donut in the far-field, and you won't be able to tell which way the phase front is rotating. It could be clockwise or anti-clockwise. In the paper you cite, that example of OAM+/- is not shown. Now, in practice, you might be able to work out by tracking the field over time (i.e. as the mode coupling changes, you might be able to tell whether and OAM is + or - based on what field it used to be recently).

Point 3 : I think you may have misunderstood the 'phase shift between polarization components' I was referring to here. If you split with a PBS, you'd be able to recover the mode weights in each polarization like you say, but you wouldn't know the phase shift between the polarizations (because they have not interfered, they're on separate photodetector arrays). e.g. imagine the simple case, where you just have a Gaussian (single mode), and you split it up and look at it on a camera in each polarization. You would know the amplitude in H and the amplitude in V, but you would not have enough information to find the polarization state. Imagine equal power in H and V, you could not distinguish between 45 degree polarization, -45, RHC, and LHC. To get around that, you'd have to do 4 measurements (rather than just 2) and do something like Stokes polarimetry. In coherent detection schemes with a local oscillator, you get around this because everything is defined relative to the phase of the local oscillator.

Reviewer #3

(Remarks to the Author)

I apologize to the authors, my initial review did not get through as I pasted it into the "Comments to the editors" section instead of the one for the authors.

Here it is:

In the current paper, the authors address the challenge of mode decomposition in multimode fibers (MMFs). MMFs represent one of the most promising avenues for increasing data rates in optical telecommunications without a proportional increase in cost. However, the problem of efficient multiplexing and demultiplexing remains an open and critical challenge.

Several hardware solutions exist, such as multiplane light converters and photonic lanterns. However, these solutions are costly and, to the best of my knowledge, have not yet met telecommunication standards regarding cross-talk and losses. Another approach involves measuring the complex field and projecting it onto a known mode basis.

This method requires a local oscillator and good interferometric stability—typically impractical for real-world implementations.

This context has spurred growing interest in intensity-based mode decomposition. Numerous methods have been proposed, often relying on capturing full intensity patterns at the fiber output combined with computationally demanding algorithms.

In this paper, the authors present a novel approach that achieves full field decomposition into a mode basis using relatively few detectors and straightforward, fast, deterministic algorithms.

The article is technical but well-structured and clearly presented. The proposed method shows strong potential as a foundation for industrial implementation, being cost-effective, relatively simple, and fast. I support the publication of this paper, provided the following comments and concerns are addressed:

Comments and Suggestions

1. Accessibility of Data and Code

A significant issue is the lack of publicly available data and code. Although the authors mention that these materials are available "upon reasonable request," I believe that for a study heavily reliant on numerical processing, public accessibility is essential.

Moreover, as the image resolution in the study is relatively low (128 × 128), hosting the dataset for public download should

not pose a significant challenge.

Ensuring data and code availability is more than a suggestion—it is a fundamental requirement for reproducibility in scientific research. Without access to the code, readers cannot independently verify the results of this work. Furthermore, the paper's impact could be greatly enhanced if the community could benchmark other state-of-the-art solutions or new techniques against these results.

2. Scalability

The authors studied the requirements and efficiency of their approach for 3- and 5-mode-fibers. A future proof solution would require the approach to be efficient and computationally reasonable when increasing the number of modes. While this paper is presented as a proof of concept, I would expect at least a discussion about the scalability, i.e. about the computational complexity and the reconstruction error when the number of modes is increased.

3. Comparison Between Experimental Results and Simulations

The paper discusses simulations to evaluate the effect of noise on mode decomposition, presenting results in Fig. 8, which show the error in mode coefficients versus the SNR. However, the location of the experimental results relative to these predictions is unclear.

What is the experimental SNR?

Additionally, while simulations use the mean absolute error (MAE) as a metric, the experimental section employs a different metric (correlation distance) to assess reconstruction quality. To provide a clear and consistent comparison, it is crucial to compare experimental and simulation results using the same metric.

4. Experimental Results and Noise Models

My guess is that the experimental results do not match the simulation predictions as closely as desired. This discrepancy may stem from the noise model used in the simulations, which considers only additive Gaussian noise.

In practice, mode decomposition errors are often influenced more by static biases than by purely stochastic noise. Common sources of such biases include optical aberrations (e.g., phase slopes or defocus) and alignment errors.

The authors address alignment errors with an optimization algorithm for the fiber's center position and magnification. A similar approach could be extended to correct simple aberrations, such as those represented by the first three Zernike polynomials: two orthogonal phase slopes and a quadratic phase (defocus).

In my opinion, applying such corrections would likely lead to improvements in reconstruction accuracy and should be investigated.

Conclusion

While the paper demonstrates considerable technical innovation and potential, addressing the aforementioned issues would strengthen the results and enhance the paper's overall impact. Specifically, ensuring data and code availability, aligning metrics between simulations and experiments, and incorporating corrections for static biases would address reproducibility concerns and elevate the quality of this work.

Version 2:

Reviewer comments:

Reviewer #2

(Remarks to the Author)

I'm satisfied with the changes that point out those few niche cases where the method wouldn't work without additional measurements.

Reviewer #3

(Remarks to the Author)

1. Accessibility of Data and Code

I thank the authors for making the data and code available, I am quite satisfied on this point.

2. Scalability

First and foremost, the noise resilience of the proposed method decreases significantly as the number of modes increases. Does that mean the techniques is not scalable? This is important to set the scope of potential applications then. While it is still interesting, it limits the applications to few-mode fibers, which should be clearly stated.

3. Comparison Between Experimental Results and Simulations

To my comment: "Additionally, while simulations use the mean absolute error (MAE) as a metric, the experimental section employs a different metric (correlation distance) to assess reconstruction quality. To provide a clear and consistent comparison, it is crucial to compare experimental and simulation results using the same metric.", the authors replied:

> Thank you for the comment. We would like to clarify that the true weight distribution of the eigenmodes is unknown to us, as determining it would require an off-axis digital holography setup or a similar technique, which we currently do not have access to. However, we would like to note that in our experimental verification, we follow established practices from multiple previous studies (e.g., refs 10, 11, 1-20, 25-31) that use intensity correlation functions to assess the accuracy of mode decomposition. This approach is widely adopted due to the complexity of setups required for independent amplitude and phase measurements, which is one reason intensity-only methods have been developed. Consequently, there is no independent wavefront measurement, and the retrieved wavefront cannot be directly compared to experimental data. That said, the full wavefront is retrieved. Since the modal weights are determined up to a constant phase shift and complex conjugation, the intensity has a one-to-one correspondence with the modal weights. A high correlation between the measured and retrieved intensity indicates that the retrieved modal weights closely match the true weights, subject to constant phase shift and complex conjugation.

I understand that, since you do not have access to a ground truth for the actual coefficient, the MAE is not accessible for the experimental data, thus you used correlation. However, from the simulations, you can compute similar curves as in Fig.8 but for the correlation and then place the experimental point(s) in the same figure to assess the agreement. So you do have a way to compare both in the same graph, if not using MAE, using correlation.

4. Experimental Results and Noise Models

The authors do not seem inclined to correct further the aberrations numerically. I understand the point of the proof of concept, that could be done with minor tweaks to the code in my opinion. Still, I will not insist further on this point.

Conclusion

I agree with the publication of the current paper, while I would prefer my last remarks to be addressed, I will not reserve my opinion on another authors response.

We sincerely thank the reviewers for their thoughtful and constructive feedback, which has helped us improve the clarity, completeness, and rigor of the manuscript. In response to the comments, we have addressed all concerns and incorporated the suggested clarifications into the manuscript.

Below, we detail our responses to each point raised.

Reviewer #1:

Sparse intensity sampling for ultrafast full-field reconstruction in low-2 dimensional photonic systems

In this article the authors demonstrate a method to reconstruct the complex field transmitted by low mode optical fibers with only intensity measurements without the need of an interferometer. The intensities are measured at a few points (sparse sampling) arranged in circular patterns.

The article is similar to a previous article on “Intensity-only-measurement mode decomposition in few-mode fibers” (E. Manuylovich et al, Opt. Exp. 29, 36769, (2021)) and the main difference is that instead of using the full image captured by the camera now it is possible to only use the measurements in a few pixels in the camera.

The main advantage of this method is that, in principle, one could build an array of fast photodiodes placed at specific coordinates on order to achieve the rates of GHz. The disadvantage is that you have to calculate those optimal positions and then build the fast photodiode array at exactly those positions that only work for a specific fiber.

1. In the abstract and the conclusion you claim that your result is 3 orders of magnitude faster than the fastest previously published result. The Opt. Exp. Paper (also claims to outperform the fastest), the mean decomposition time is 52 microseconds (Caption of Fig. 15). I understand that the 3 orders of magnitude refer to the number of multiplications, which might not be a 3 orders of magnitude in the processing time?

Thank you for bringing this to our attention. Indeed, the relationship between computational complexity and processing time is influenced by several factors, including the architecture of the computing device, the number of operations it can perform per cycle, memory bandwidth limitations, computational overhead, and others. While processing time is not always directly proportional to the number of multiplications, it is often used as a rule of thumb for estimating performance. However, we acknowledge that a more detailed discussion of these factors would provide greater clarity.

2. Figures 1, 5 and 6 show complex images (intensity and phase), but there is no colorbar or explanation of the colorcode. Also, what is the size of the images in terms of pixels in the camera (and pixel size)?

Thank you for your comment. We appreciate your feedback. The captions for Figures 1, 5, and 6 have been updated to explicitly clarify the meaning of the color code, including the fact that phase is represented by color and spans a range from 0 to 2π . We have also clarified the image size in terms of pixels before Figure 5, noting that these figures correspond to simulation results. As such, the pixel size is not fixed and would depend on experimental conditions. The overall scale of these figures, if measured experimentally, would vary based on the parameters of the lenses used to project the intensity patterns onto the camera.

3. The experimental results of Fig. 11 only show the intensity distributions. What about the phase distributions?

In our experimental verification, we follow established practices from multiple previous studies (e.g., refs 10, 11, 18-20, 25-31) that use intensity correlation functions to assess the accuracy of mode decomposition. This approach is widely adopted due to the complexity of setups required for independent amplitude and phase measurements, which is one reason intensity-only methods have been developed. Consequently, there is no independent wavefront measurement, and the retrieved wavefront cannot be directly compared to experimental data. That said, the full wavefront is retrieved. Since the modal weights are determined up to a constant phase shift and complex conjugation, the intensity has a one-to-one correspondence with the modal weights. A high correlation between the measured and retrieved intensity indicates that the retrieved modal weights closely match the true weights, subject to constant phase shift and complex conjugation.

4. Also, it seems that right now your experiment is limited to a maximum of 5 modes. Is this also true for simulations? You mention the symmetries in the intensity patterns, which would appear to be a fundamental limitation? Or what is the mode number limit for your simulations?

The experimental verification is conducted on 3-mode fibers. For simulations, we tested the method on fibers with 3 to 10 modes, with results for 3- and 5-mode fibers included in the paper. The main limitation of the method is its robustness to noise. Specifically, the method involves multiplying the measured intensity vector I_p by the matrix $\Xi = \mathbf{G}^{-1}(\mathbf{P}\mathbf{Y}_\perp)^\dagger$, and here, the orthogonalization matrix \mathbf{G} has a condition number that increases with the number of modes. This leads to a higher condition number for the overall matrix Ξ , and this is what we discover in table 2 in the manuscript. This is the main limitation of the proposed method from performing mode decomposition in fibers with a number of modes higher than 5 when any reasonable amount of noise is added. This is because matrix multiplication by a matrix with a given condition number amplifies noise by this condition number (see, for example, Johnson and Horn, *Matrix analysis*, Cambridge University Press, Cambridge, 1985, p. 336). In simulations, the method becomes infeasible for more than approximately 48 modes, as demonstrated in Manuylovich et al., *Optics Express*,

29(22), 36769 (2021). This corresponds to the precision limit of double-precision numbers: when the condition number exceeds the inverse machine epsilon for double-precision numbers, numerical instability arises.

Another important consideration is that the number of required photodetectors scales quadratically with the number of modes. Therefore, we believe the proposed method is most advantageous for 3- and 5-mode systems.

Reviewer #2:

Overall, I think the technique could have some utility, but I don't think it would be particularly useful for use directly in mode division multiplexing, which appears to be the emphasis in the paper. I believe some of the other areas mentioned in the paper such as general beam characterization and few-mode fiber sensing would be more promising.

I'll start here by listing some problems I see for using the system directly in mode division multiplexing, I'll get these negative points out of the way first as there's other potentially better applications I can think of which I'll discuss in a moment.

Downsides of the method for direct use as mode demultiplexer...

1. It's quite lossy, the photodetectors only sample a small portion of the beam.

We acknowledge this point. Indeed, the photodetectors sample only a small portion of the beam, which is a limitation of the method. This drawback is particularly significant for long-haul and metro transmission. However, for short-reach networks, such as access or data-center networks, this limitation may be less critical when weighed against the simplicity and practicality of the detection method.

2. It can't distinguish between conjugate versions of the beam. The paper mentions limiting the modulation range to get around this, but presumably at the end of a fiber, with unknown mode coupling, you could end up with a scenario along the lines of having to distinguish between an OAM +1 mode, and the conjugate OAM -1, and have no ability to tell the difference. Also, limiting the modulation range would limit the information capacity of the link, and would probably not be very popular unless there's a significant upside that comes along with it.

We acknowledge this point. The proposed basic implementation is indeed unable to distinguish between conjugate fields. However, this limitation can be addressed with a simple modification: taking an additional intensity measurement in another plane. A similar approach is described in our previous work (Manuylovich et al., Optics Express, 29(22), 36769, 2021), where we demonstrated that measuring both near- and far-field intensities

enables full wavefront retrieval without ambiguity between complex conjugates. A corresponding note (“Alternatively, this ambiguity can be fully resolved by combining intensity measurements from different planes...” above Eq. 3) has been added to the manuscript for clarification.

3. How could your technique be used along with polarization? For mode division multiplexing you would need the ability to decipher the phase shift between polarization components. It would typically also be desirable to use wavelength division multiplexing as well.

Thank you for raising this point. To detect signals with polarization multiplexing, the two polarizations can be separated using components such as a polarizing beam splitter (PBS), with each polarization directed to a separate group of photodetectors. This approach is well-established in coherent receivers, as described in works like doi.org/10.1364/OE.19.00B112, where a dual-polarization optical hybrid is depicted in Fig. 4. In our work, we demonstrate how mode weights can be retrieved for single polarization. For dual-polarization signals, the polarizations would need to be split, and the mode retrieval process would need to be applied independently to each polarization. This is analogous to the techniques used in optical communications with single-mode fibers. A note clarifying this (“For dual-polarization signals, polarization components would need to be separated, for example, using a polarizing beam splitter, with the mode retrieval process applied independently to each polarization”) has been added to the manuscript, to the Methods section.

4. I suspect your scheme cannot reconstruct multiple fields at once? You’ve shown nice examples of recovering a single field, but for mode division multiplexing, the output beam would consist of multiple fields corresponding with multiple independent information channels, all with their own amplitude and phase modulations over the top. To be true mode demultiplexer you would have to be able to recover them all at once.

Thank you for your question. If I understand correctly, you are asking about the ability to reconstruct multiple spatial modes (fields) simultaneously, as required for mode division multiplexing (MDM). In MDM, different spatial modes serve as parallel information channels, with each mode's amplitude and phase modulated independently to encode data streams using modulation formats like QAM16 (amplitude + phase) or PAM4 (amplitude only). To decode these streams, it is necessary to retrieve the amplitudes and phases of individual spatial modes. This process (aka mode decomposition) is what we address in this work. Specifically, we demonstrate that all amplitudes and phases of multiple spatial modes can be decoded simultaneously using only quadratic photodetectors, even in the presence of noise with reasonable SNR. Figure 6 provides an example of this, showing the spatial field distribution in a 5-mode fiber, where mode

amplitudes and phases are retrieved and displayed (see the subfigure on the right-hand side that shows encoded and retrieved mode weights). For few-mode fibers, the spatial distribution of the electromagnetic field is fully characterized by the eigenmode weights (by the definition of eigenmodes). Thus, all independent spatial channels (with their amplitude and phase modulations) are decoded at once in our approach. If your comment refers to different carrier wavelengths (wavelength division multiplexing), that is outside the scope of this work and is not considered here. I apologize if I misunderstood your question.

There has also been work on 'High-dimensional Stokes receivers' by groups such as William Shieh at University of Melbourne and Joseph Kahn at University of Stanford

<https://doi.org/10.1364/OL.44.002065>

That can reconstruct multiple channels at once, with no local oscillator (external reference), at telecommunications rates (GHz). These use more photodetectors than your approach, but these can also unambiguously recover multiple channels.

Thank you for pointing this out. The referenced work employs a photonic lantern for spatial mode demultiplexing (see MSPL in Fig. 1 and its caption). This technique, along with multi-plane light conversion (MPLC), is among the most common methods for physically separating spatial modes and coupling them into single-mode fibers. In the referred study, the authors use these separated modes and mix them through multiple couplers, including fancy 3×3 couplers, to retrieve mode weights. In contrast, our approach does not rely on complex 3D-engineered waveguides like photonic lanterns or free-space devices like MPLC. This simplicity eliminates the need for complex and expensive hardware (MPLCs for 6 modes are ~6000 USD), which we believe is a significant advantage of our method. The cited paper has been added to the manuscript, replacing the reference "14. Birks, T. A., Gris-Sánchez, I., Yerolatsitis, S. et al. The photonic lantern. *Adv. Opt. Photonics* 7, 107–167 (2015)". While the original reference introduced the concept of photonic lanterns, the new citation, provided by you, focuses on their application for mode decomposition.

Potentially better application could be few-mode fiber sensing?

It might be more useful, not so much for telecoms, but as a general characterization or sensing tool. Perhaps in some application like sensing where you want to measure the drift of the modal content over time to discern some property (such as strain or temperature?), that drift might even help to distinguish between ambiguities that you can't otherwise resolve. For example, the difference between an OAM +1 mode, and an OAM -1 mode might be clear due to the modal content that preceded it in time? In that case you might only want a few modes, and you might want the high-speed aspect without going to a full-on coherent detection scheme? The simplicity of a few well placed

photodetectors and high-speed reconstruction could be useful there. Perhaps it would also be useful for scenarios at wavelength where coherent detection isn't convenient or possible? i.e. not at 1310/1550nm. Just as you've done in this work where you've demonstrated at 650nm.

Thank you for this insightful suggestion. We agree that the proposed method could have potential applications beyond telecommunications, such as in few-mode fiber sensing for strain or temperature measurements, as you described. The simplicity and high-speed reconstruction aspects of our approach could indeed make it suitable for scenarios where coherent detection is impractical, such as at wavelengths outside the traditional telecom range (e.g., 1310/1550 nm). For example, at 850 nm, which is currently heavily used for short-reach data center interconnects. While the focus of our current work is on the demonstration of the technique and its potential, exploring applications could be an interesting direction for future research.

This is just an observation, but it looks like the results of Fig. 3, are similar/same to those found for attempting to find the optimal placing of spot couplers for N=6 fibers in coherent MDM. The mode basis looks the same, and the spot placement looks the same.

e.g. <https://ieeexplore.ieee.org/abstract/document/6317139>

Thank you for your observation. These are also pretty similar to the arrangement that can be found in photonic lanterns. This was a surprising thing to discover. However, the important difference is that we have 6 photodetectors for 3-mode fiber (<https://ieeexplore.ieee.org/abstract/document/6317139> or photonic lanterns use 3 spots for 3-mode fibers). Photonic lanterns always use M-spot patterns for M-mode demultiplexers. We utilize $M(M+1)/2$ quadratic photodetectors M-mode fibers, for example, 6 photodetectors for 3-mode fibers and 15 photodetectors for 5-mode fibers. But for a given number of spots of photodetectors, the patterns of their placements look very similar. We believe that this is rooted in the type of symmetry and the number of zeroes for the highest-order Bessel function that enters eigenmodes of a particular fiber (when we consider weakly guiding step-index fibers).

2.3 : I was confused by the difference between near-field and far-field measurements. I don't know if you mention anywhere what type of fiber/modes you're working with? Is it step-index? If it's step-index I understand how there would be a difference between near and far-field, for graded-index (parabolic), I'd expect the near and far-fields to be identical. Later on in Section 3, you mention that it's step-index, but mentioning that somewhere earlier for the simulations of Section 2 would be helpful.

Thank you for your comment. We have amended the manuscript to address this point. At the beginning of the Results section, we now include a refined statement: "...we consider few-mode optical fibers with a step-index profile in the LP mode approximation..."

Am I correct in assuming that tests such as Fig. 5 are generated numerically (i.e. these aren't measured on real cameras etc?), it seems from later in the document that it is simulated, but it would be good to clarify earlier. And then in Section 3, it becomes experimental results?

Thank you for your comment. When referring to Figures 5 and 6, we have mentioned that “simulation results shown on 64×64 pixel grids”. However, to clarify further and avoid any ambiguity, we have revised the text to explicitly state: “Figures 5 and 6 provide examples of corresponding sparse sampling mode decomposition in 3- and 5-mode fibers, respectively. These results were obtained through numerical simulations on 64×64 pixel grids. Later in Section 3, we present experimental results.”

Minor : the intensity plots of Figure 9 might be better as simply traditional 2D plots rather than 3D, it'd be easier to see the features in 2D.

Thank you for your comment. The main purpose of the 3D plot in Fig. 9 is to highlight the experimentally measured intensity distributions, which show a non-zero “noisy pedestal” outside the main beam area. This is a key aspect we needed to account for when processing the experimental data. The 3D representation, along with the smaller inset on the left-hand side of Fig. 9, was chosen to emphasize this pedestal rather than the intensity distribution itself. We have amended the caption to Fig. 9 to clarify this point.

In your 4. Discussion and Conclusion, you state “in any photonic system in which the feature space of an electromagnetic field is highly degenerate or inherently low-dimensional”, I think you should change that to be “in other photonic systems”, i.e. I don't think it'd work for “any” low-order photonic system. For example, an array of single mode waveguides? E.g. 3 single-mode waveguides is a 3 mode system, but I don't think your system could handle that scenario because the individual single mode waveguides don't occupy any of the same space, so you can't measure their interference. Although perhaps it could still work if you look at it in the far-field where the waveguides can interfere?

Thank you for this comment. The main requirement is that the matrix of pairwise mode products must be well-conditioned, so modes need to be overlapping in the measurement plane so that for each pair Ψ_k and Ψ_j there are (x, y) such that $\Psi_k(x, y)\Psi_j(x, y) \neq 0$. Indeed, the proposed method can only be applied to low-dimensional photonic systems, where we are able to observe the interference between different modes. So for your example with multi-core fibers, one indeed needs to measure the intensity pattern in a measurement plane, in which the modes overlap, for example, far field. We added the clarification to the Discussion section (“It only requires that the modes exhibit non-zero overlap in the measurement plane to ensure that the matrix of pairwise mode products is well-conditioned”).

Minor (e.g. typos)...

Abstract : “have too high a computational complexity to be applicable at high {word missing, presumably ‘speed’?}”

Typo, Fig 4 “InIntensity”

Thank you for pointing this out. We corrected these typos and also went through the manuscript thoroughly to find and correct any other typos.

We sincerely thank all the reviewers for their constructive feedback and valuable suggestions. Their insightful comments have helped us significantly improve the clarity and quality of our manuscript. Below, we address each reviewer's comments in detail, highlighting the revisions made in the manuscript and clarifying specific aspects where necessary.

Reviewers' comments:

Reviewer #1 (Remarks to the Author):

The authors answered my comments and clarified the points that I raised.

We would like to thank the reviewer for the constructive feedback that helped to improve the quality and clarity of the manuscript.

Reviewer #2 (Remarks to the Author):

2. "The proposed basic implementation is indeed unable to distinguish between conjugate fields. However, this limitation can be addressed with a simple modification: taking an additional intensity measurement in another plane."

I don't think this is the case specifically for OAM ± 1 modes. My understanding is you could use the two plane approach for almost all fields, except for a few specific cases where it fails. Namely, scenarios where two modes (e.g. OAM+ and OAM-) both have identical near-fields and far-fields. You can't tell the difference between those two orthogonal modes, because they have the property of looking the same in both planes. i.e. you'll see a donut in the near-field, and a donut in the far-field, and you won't be able to tell which way the phase front is rotating. It could be clockwise or anti-clockwise. In the paper you cite, that example of OAM \pm is not shown. Now, in practice, you might be able to work out by tracking the field over time (i.e. as the mode coupling changes, you might be able to tell whether and OAM is + or - based on what field it used to be recently).

Thank you for your comment. We agree that for OAM modes with $l=\pm 1$, the two-plane intensity measurement approach may not be sufficient to distinguish between conjugate fields, as both modes exhibit identical near-field and far-field intensity profiles. More importantly, their corresponding field distributions are complex conjugates of each other. To address this, we have expanded the discussion on the limitations of our proposed method. Specifically, we now note that the method, in its current form, is not applicable to low-dimensional photonic systems where some modes are complex conjugates of one another. We also clarify that when the modes are not linearly polarized, they are not their own complex conjugates, and the method requires more measurements. In particular, M^*M measurements are enough to reconstruct mode weights fully. And there is no limitation on complex conjugation as long as neither of the modes is a complex conjugate of some other mode (other than itself). We added the clarification to the paper. In light of the previous discussion on "low-dimensional systems," particularly in the context of multi-core systems and the applicability of our method to multi-core fibers, we would like to highlight a recent paper in Optics Letters (<https://doi.org/10.1364/OL.550270>). This study employs a similar (albeit non-sparse) approach for mode decomposition in the far field: essentially the same strategy (using far-field interference) that we discussed in the last review round. We have now cited this paper in our manuscript.

Point 3 : I think you may have misunderstood the 'phase shift between polarization components' I was referring to here. If you split with a PBS, you'd be able to recover the mode weights in each polarization

like you say, but you wouldn't know the phase shift between the polarizations (because they have not interfered, they're on separate photodetector arrays). e.g. imagine the simple case, where you just have a Gaussian (single mode), and you split it up and look at it on a camera in each polarization. You would know the amplitude in H and the amplitude in V, but you would not have enough information to find the polarization state. Imagine equal power in H and V, you could not distinguish between 45 degree polarization, -45, RHC, and LHC. To get around that, you'd have to do 4 measurements (rather than just 2) and do something like Stokes polarimetry. In coherent detection schemes with a local oscillator, you get around this because everything is defined relative to the phase of the local oscillator.

Thank you for your clarification. We agree that the proposed method, without modifications, cannot be used to directly recover the phase shift between polarization states. As you pointed out, splitting with a PBS allows us to determine the mode weights in each polarization, but it does not provide information about the relative phase shift. However, we would like to note that once the mode weights in both polarizations are known, an additional intensity measurement at the same sparse spots (this time without polarization separation) allows for the recovery of the polarization state's ellipticity. This enables determining the phase shift between different polarization components, albeit up to $\text{mod}(\pi)$. We have added this clarification to the discussion section. That said, we believe a detailed exploration of this aspect falls outside the scope of this manuscript.

Reviewer #3 (Remarks to the Author):

I apologize to the authors, my initial review did not get through as I pasted it into the "Comments to the editors" section instead of the one for the authors.

Here it is:

In the current paper, the authors address the challenge of mode decomposition in multimode fibers (MMFs). MMFs represent one of the most promising avenues for increasing data rates in optical telecommunications without a proportional increase in cost. However, the problem of efficient multiplexing and demultiplexing remains an open and critical challenge.

Several hardware solutions exist, such as multiplane light converters and photonic lanterns. However, these solutions are costly and, to the best of my knowledge, have not yet met telecommunication standards regarding cross-talk and losses.

Another approach involves measuring the complex field and projecting it onto a known mode basis.

This method requires a local oscillator and good interferometric stability—typically impractical for real-world implementations.

This context has spurred growing interest in intensity-based mode decomposition. Numerous methods have been proposed, often relying on capturing full intensity patterns at the fiber output combined with computationally demanding algorithms.

In this paper, the authors present a novel approach that achieves full field decomposition into a mode basis using relatively few detectors and straightforward, fast, deterministic algorithms.

The article is technical but well-structured and clearly presented. The proposed method shows strong potential as a foundation for industrial implementation, being cost-effective, relatively simple, and fast. I support the publication of this paper, provided the following comments and concerns are addressed:

We sincerely thank the reviewer for their thoughtful and detailed feedback, as well as for their support and positive assessment of our work. We appreciate the recognition of the potential industrial relevance of our approach and the encouraging remarks regarding its cost-effectiveness, simplicity, and speed. We carefully reviewed the comments and concerns provided and will address each point in detail below to ensure clarity and improve the manuscript further.

Comments and Suggestions

1. Accessibility of Data and Code

A significant issue is the lack of publicly available data and code. Although the authors mention that these materials are available "upon reasonable request," I believe that for a study heavily reliant on numerical processing, public accessibility is essential.

Moreover, as the image resolution in the study is relatively low (128×128), hosting the dataset for public download should not pose a significant challenge.

Ensuring data and code availability is more than a suggestion—it is a fundamental requirement for reproducibility in scientific research. Without access to the code, readers cannot independently verify the results of this work. Furthermore, the paper's impact could be greatly enhanced if the community could benchmark other state-of-the-art solutions or new techniques against these results.

We appreciate the reviewer's emphasis on the importance of data and code accessibility for ensuring reproducibility and fostering further research. We fully agree that sharing these resources is crucial for scientific transparency and enabling the community to build on our work.

In response, we have published raw experimental data and the near-field implementation of the proposed approach on GitHub: <https://github.com/egor-manu/sparse-intensity-sampling-ultrafast-full-field-reconstruction>. We have also updated the manuscript to reflect this, adding the following statement under the "Availability of Data and Materials" section: "The raw experimental dataset and the basic implementation of the proposed approach for near-field measurements are available at <https://github.com/egor-manu/sparse-intensity-sampling-ultrafast-full-field-reconstruction>".

We hope this addition addresses the reviewer's concern and supports the broader community in validating and extending our work.

2. Scalability

The authors studied the requirements and efficiency of their approach for 3- and 5-mode-fibers. A future proof solution would require the approach to be efficient and computationally reasonable when increasing the number of modes. While this paper is presented as a proof of concept, I would expect at least a discussion about the scalability, i.e. about the computational complexity and the reconstruction error when the number of modes is increased.

Thank you for the insightful comment regarding scalability. There are two primary reasons why we focused on 3- and 5-mode fibers in this study.

First and foremost, the noise resilience of the proposed method decreases significantly as the number of modes increases. We have added the following explanation to the discussion section:

"The proposed method faces challenges with mode decomposition in fibers supporting six or more modes due to symmetries in the intensity patterns. These symmetries increase the condition number $\kappa(\Xi)$, leading to significant noise amplification during the inference stage. For example, in a six-mode fiber in the far-field plane, the condition number of the full matrix Y is $\kappa(Y) \approx 200$, which corresponds to a noise amplification of approximately 23 dB. This results in substantial reconstruction errors, rendering the method impractical for fibers with a higher number of modes."

The second reason is that the number of required photodetectors scales quadratically with the number of modes. Even if the condition number were not a limiting factor, the rapid increase in the number of photodetectors would pose a practical challenge. However, the primary limitation remains noise amplification due to the increased condition number $\kappa(\Xi)$.

We appreciate your suggestion and hope this discussion clarifies the scalability limitations of our current approach.

3. Comparison Between Experimental Results and Simulations

The paper discusses simulations to evaluate the effect of noise on mode decomposition, presenting results in Fig. 8, which show the error in mode coefficients versus the SNR. However, the location of the experimental results relative to these predictions is unclear.

What is the experimental SNR?

We added the estimated experimental SNR in the caption to Fig. 9. It now says "The estimated noise level in the experiment is 18.5."

Additionally, while simulations use the mean absolute error (MAE) as a metric, the experimental section employs a different metric (correlation distance) to assess reconstruction quality. To provide a clear and consistent comparison, it is crucial to compare experimental and simulation results using the same metric.

Thank you for the comment. We would like to clarify that the true weight distribution of the eigenmodes is unknown to us, as determining it would require an off-axis digital holography setup or a similar technique, which we currently do not have access to. However, we would like to note that in our experimental verification, we follow established practices from multiple previous studies (e.g., refs 10, 11, 18-20, 25-31) that use intensity correlation functions to assess the accuracy of mode decomposition. This approach is widely adopted due to the complexity of setups required for independent amplitude and phase measurements, which is one reason intensity-only methods have been developed. Consequently, there is no independent wavefront measurement, and the retrieved wavefront cannot be directly compared to experimental data. That said, the full wavefront is retrieved. Since the modal weights are determined up to a constant phase shift and complex conjugation, the intensity has a one-to-one correspondence with the modal weights. A high correlation between the measured and retrieved intensity

indicates that the retrieved modal weights closely match the true weights, subject to constant phase shift and complex conjugation.

4. Experimental Results and Noise Models

My guess is that the experimental results do not match the simulation predictions as closely as desired. This discrepancy may stem from the noise model used in the simulations, which considers only additive Gaussian noise.

In practice, mode decomposition errors are often influenced more by static biases than by purely stochastic noise. Common sources of such biases include optical aberrations (e.g., phase slopes or defocus) and alignment errors.

The authors address alignment errors with an optimization algorithm for the fiber's center position and magnification. A similar approach could be extended to correct simple aberrations, such as those represented by the first three Zernike polynomials: two orthogonal phase slopes and a quadratic phase (defocus).

In my opinion, applying such corrections would likely lead to improvements in reconstruction accuracy and should be investigated.

Indeed, the discrepancies between the measured and retrieved images contain systematic errors, as seen in the lower row of Fig. 11. These errors likely arise from imperfections in the optical system, including misalignment and lens sphericity in the 4-f imaging setup. We addressed basic alignment and scale corrections to demonstrate the experimental feasibility of the proposed approach. Improving mode decomposition accuracy through additional aberration correction or a more advanced optical system is certainly possible. However, the main goal of this experiment was to demonstrate the feasibility of the proposed method, and such improvements are beyond the scope of this work. We agree that these corrections could enhance reconstruction accuracy and should be explored in future studies.

Conclusion

While the paper demonstrates considerable technical innovation and potential, addressing the aforementioned issues would strengthen the results and enhance the paper's overall impact. Specifically, ensuring data and code availability, aligning metrics between simulations and experiments, and incorporating corrections for static biases would address reproducibility concerns and elevate the quality of this work.

We believe that we have thoroughly addressed all the concerns and suggestions provided by the reviewers. Our manuscript has been updated accordingly to incorporate the requested clarifications and improvements. We trust that the revisions enhance the scientific rigor, clarity, and overall impact of our work. We are grateful for the reviewers' efforts and thoughtful feedback and look forward to the final evaluation.

Response to the reviewers' comments

We thank the reviewers for their thorough evaluation of our manuscript and their constructive feedback. We are pleased that the revisions and clarifications we provided have addressed most of the concerns. Below, we respond point by point to the remaining comments, noting the changes made and clarifying our reasoning where necessary. We are grateful for the reviewers' thoughtful engagement, which has helped improve the clarity and quality of the manuscript.

REVIEWERS' COMMENTS:

Reviewer #2 (Remarks to the Author):

I'm satisfied with the changes that point out those few niche cases where the method wouldn't work without additional measurements.

We sincerely thank the reviewer for their positive feedback. We're glad the clarifications regarding the method's limitations in niche cases were satisfactory. We appreciate the reviewer's careful reading and helpful suggestions throughout the revision process.

Reviewer #3 (Remarks to the Author):

1. Accessibility of Data and Code

I thank the authors for making the data and code available, I am quite satisfied on this point.

We thank the reviewer for acknowledging the availability of our data and code. We're pleased to hear that this aspect is satisfactory and appreciate the reviewer's support.

2. Scalability

First and foremost, the noise resilience of the proposed method decreases significantly as the number of modes increases.

Does that mean the techniques is not scalable? This is important to set the scope of potential applications then. While it is still interesting, it limits the applications to few-mode fibers, which should be clearly stated.

We thank the reviewer for this valuable observation. We clarify in the Discussion that while the method performs well for 3- and 5-mode fibers, it faces limitations for fibers with six or more modes due to increased symmetry in intensity patterns, which leads to higher condition numbers and noise amplification. This constrains the method's scalability but still allows efficient mode decomposition in the few-mode fibers with 3 orders of magnitude

lower computational complexity than the fastest previous method, and 6-7.5 orders of magnitude higher speed than CNN-based approaches.

3. Comparison Between Experimental Results and Simulations

To my comment: “Additionally, while simulations use the mean absolute error (MAE) as a metric, the experimental section employs a different metric (correlation distance) to assess reconstruction quality. To provide a clear and consistent comparison, it is crucial to compare experimental and simulation results using the same metric.”, the authors replied:

> Thank you for the comment. We would like to clarify that the true weight distribution of the eigenmodes is unknown to us, as determining it would require an off-axis digital holography setup or a similar technique, which we currently do not have access to. However, we would like to note that in our experimental verification, we follow established practices from multiple previous studies (e.g., refs 10, 11, 1-20, 25-31) that use intensity correlation functions to assess the accuracy of mode decomposition. This approach is widely adopted due to the complexity of setups required for independent amplitude and phase measurements, which is one reason intensity-only methods have been developed. Consequently, there is no independent wavefront measurement, and the retrieved wavefront cannot be directly compared to experimental data. That said, the full wavefront is retrieved. Since the modal weights are determined up to a constant phase shift and complex conjugation, the intensity has a one-to-one correspondence with the modal weights. A high correlation between the measured and retrieved intensity indicates that the retrieved modal weights closely match the true weights, subject to constant phase shift and complex conjugation.

I understand that, since you do not have access to a ground truth for the actual coefficient, the MAE is not accessible for the experimental data, thus you used correlation. However, from the simulations, you can compute similar curves as in Fig.8 but for the correlation and then place the experimental point(s) in the same figure to assess the agreement. So you do have a way to compare both in the same graph, if not using MAE, using correlation.

We thank the reviewer for this helpful suggestion. We have now followed the recommendation and included correlation-based curves in Figure 8, computed from simulations. This allows for a direct comparison between simulation and experimental results using the same metric. We believe this addition strengthens the consistency of the evaluation and improves clarity.

4. Experimental Results and Noise Models

The authors do not seem inclined to correct further the aberrations numerically. I understand the point of the proof of concept, that could be done with minor tweaks to the code in my opinion. Still, I will not insist further on this point.

We appreciate the reviewer's understanding and their acknowledgment of the proof-of-concept nature of the current work. We agree that numerical correction of aberrations could be incorporated with relatively minor code adjustments and see this as a promising direction for future improvement, though we did not pursue it in the current version.

Conclusion

I agree with the publication of the current paper, while I would prefer my last remarks to be addressed, I will not reserve my opinion on another authors response.

We thank the reviewer for their overall support of the manuscript and for the constructive feedback throughout. We appreciate their thoughtful remarks and consideration.